# Uncovering the Computational Ingredients of Human-Like Representations in LLMs

## Abstract

The ability to translate diverse patterns of inputs into structured patterns of behavior has been thought to rest on both humans' and machines' ability to learn robust representations of relevant concepts. The rapid advancement of transformer-based large language models (LLMs) has led to a diversity of computational ingredients — architectures, fine tuning methods, and training datasets among others — but it remains unclear which of these ingredients are most crucial for building models that develop human-like representations. Further, most current LLM benchmarks are not suited to measuring representational alignment between humans and models, making existing benchmark scores unreliable for assessing if current LLMs are making progress towards becoming useful cognitive models. Here, we address these limitations by first evaluating a set of over 77 models that widely vary in their computational ingredients on a triplet similarity task, a method well established in the cognitive sciences for measuring human conceptual representations, using concepts from the THINGS database. Comparing human and model representations, we find that models that undergo instruction-finetuning and which have larger dimensionality of attention heads are among the most human aligned. We also find that factors such as choice of activation function, multimodal pretraining, and parameter size have limited bearing on alignment. Correlations between alignment scores and scores on existing benchmarks reveal that while some benchmarks (e.g., BigBenchHard) are better suited than others (e.g., MUSR) for capturing representational alignment, no existing benchmark is capable of fully accounting for the variance of alignment scores, demonstrating their insufficiency in capturing human-AI alignment. Taken together, our findings help highlight the computational ingredients most essential for advancing LLMs towards models of human conceptual representation and address a key benchmarking gap in LLM evaluation.

## 1 Introduction

The success of deep neural network models on diverse domains ranging from perception (Yamins et al., 2014; Yamins & DiCarlo, 2016), to robotic manipulation (Finn et al., 2016), to language understanding (Tuckute et al., 2024) has partly been attributed to their ability to learn powerful *representations* that can effectively aid in translating inputs into appropriate outputs. The continued development of a specific class of these systems — transformer-based large language models (LLMs) — and their capabilities on naturalistic human tasks might imply that these models, through training on large enough corpora of text (and often image) data and with the correct architectural ingredients, come to possess representations that are largely isomorphic to humans' mental representations. This implication is critical for many domains of cognitive science that have long sought to characterize the nature of human conceptual representations and the operations performed on them that lead to naturalistic behavior. Indeed, there is a rich tradition of using deep neural networks as *computational cognitive models*: using their learned representations as proxies for human representations, and often mapping the evolution of learned representations to development of human conceptual representations (Jackson et al., 2022; Warstadt et al., 2025). While a patchwork of evidence in various domains has shown concordances in representations between language models and people's behaviors and patterns in their neural activity, it remains unclear what properties of current LLMs are most predictive of strong human-model alignment. That

is, given the engineering-centric drive behind current LLM development, it is difficult to isolate what *computational ingredients* (model architecture, model size, instruction-tuning, activation functions, etc.) are important for giving rise to human-aligned conceptual representations in models for a large variety of concepts.

Identifying the computational ingredients underlying human-model alignment is critical for several reasons. While frontier models have achieved remarkable performance on diverse benchmarks spanning scientific reasoning (Rein et al., 2024; Suzgun et al., 2022), software engineering (Chen et al., 2021; Jimenez et al., 2023), and chart and humor understanding (Masry et al., 2022; Methani et al., 2020; Mukherjee et al., 2025; Zhou et al., 2025; Hessel et al., 2022) (domains thought to require 'human-like' thinking), we lack a unified theory explaining *why* model performance on one task might transfer to another and reasons for inconsistencies in benchmark rankings for the same model. Recent work has shown that optimizing for benchmark accuracy alone is insufficient for building models that fail in human-like ways and that are generally aligned with humans (Fel et al., 2022; Ying et al., 2025). We argue that prioritizing representational alignment (Sucholutsky et al., 2023; Sucholutsky & Griffiths, 2023b; Muttenthaler et al., 2024) offers a promising path forward. Measuring the similarity between model and human internal representations can provide a more unified account of model capabilities, guide the development of more robustly aligned systems (Collins et al., 2024; Muttenthaler et al., 2024; Peterson et al., 2018), and accelerate cognitive science by yielding more faithful computational models of the human mind.

## 2 RELATED WORK

**Quantifying Human and Model Representations using Large-Scale Concept Datasets** Mapping the geometry human conceptual knowledge has long been a central goal of cognitive science (Rogers & McClelland, 2004; Shepard, 1980; Tversky, 1977; Rumelhart et al., 1986). Early attempts to do so relied on explicit feature ratings and pairwise similarity judgments, which scaled poorly limiting their utility in capturing the structure of the vast inventory of human concepts (Rosch, 1975; Nosofsky, 1984; Shepard, 1980). While structured probabilistic models have been useful in explaining aspects of concept learning, they too have been limited often being restricted to specific relational structures, limiting their scope and generalizability (Zhao et al., 2024; Wong et al., 2022; Kemp & Tenenbaum, 2008). Recent years have seen a resurgence of similarity-based measures of characterizing human conceptual knowledge, driven by scalable algorithms that convert human judgments into expressive high-dimensional *semantic embeddings* (King et al., 2019; Mukherjee & Rogers, 2025; Hebart et al., 2019; Suresh et al., 2023b; Peterson et al., 2018; Kriegeskorte & Mur, 2012; Cichy et al., 2019; Hebart et al., 2023; Muttenthaler et al., 2022b; Jamieson et al., 2015; Sievert et al., 2023) capable of expressing latent dimensions organizing human semantic knowledge. Specifically, triadic similarity judgment tasks — where participants select the most similar item from a triplet—are a particularly powerful paradigm (Tamuz et al., 2011; Wah et al., 2014; Kleindessner & von Luxburg, 2014). Embeddings derived from this measure have shown to be good predictors of both neural and behavioral patterns in humans across a variety of semantic tasks (Mukherjee & Rogers, 2025; Mukherjee et al.; Hebart et al., 2023; Colon & Rogers, 2023; Suresh et al., 2023b; Marjieh et al., 2022; Mukherjee et al., 2022; Mur et al., 2013; Suresh et al., 2024).

A complementary advancement in recent years that has galvanized research in representational alignment is the development of concept datasets that aim to capture the diversity of object concepts humans reason about (Mehrer et al., 2021; Hebart et al., 2019; Giallanza et al., 2024; Suresh et al., 2025). Datasets like THINGS (Hebart et al., 2019) and ECOSET (Mehrer et al., 2021) consist of concept sets and associated images along with rich psychological and neural metadata (Hebart et al., 2023) that allow for rich comparisons between human and model representations along multiple levels of analysis — behavioral, neural, and computational.

**Representational Alignment: Measurement and Downstream Task Performance** The nascent field of representational alignment has developed formal tools for comparing internal representations across biological and artificial systems (Kriegeskorte et al., 2008; Sucholutsky et al., 2023; Barbosa et al., 2025; Oswal et al., 2016). Core measurement techniques include Representational Similarity Analysis (RSA) using correlation between similarity matrices derived from behavioral, neural, or neural network activation data (Kriegeskorte et al., 2008; Nili et al., 2014), Procrustes analysis for finding optimal linear alignments (Schönemann, 1966), and Centered Kernel Alignment (CKA)

for comparing representations invariant to linear transformations (Kornblith et al., 2019; Williams et al., 2021). Recent work has further developed more sensitive metrics including shape-based metrics (Williams et al., 2021), topological measures (Barannikov et al., 2021), information-theoretic approaches (Bansal et al., 2021), and regularized regression-based methods Oswal et al. (2016) each of which iteratively address issues relating to model regularization, generalization, and error-bounds . Growing evidence demonstrates that representational alignment with humans correlates with practical benefits for model performance. Models with higher human alignment show improved few-shot learning (Sucholutsky & Griffiths, 2023a; Huh et al., 2024), better out-of-distribution generalization (Moschella et al., 2023; Norelli et al., 2023), enhanced adversarial robustness (Engstrom et al., 2019), and more human-like error patterns (Geirhos et al., 2021; Fel et al., 2022). These findings suggest that human-aligned representations capture meaningful invariances that support robust intelligence.

**Computational Ingredients Driving Alignment** Understanding which design choices yield human-aligned representations in foundation models can help uncover general principles undergirding human intelligence. For instance, in vision, early work showed that optimizing models for categorization (Yamins et al., 2014) and later for contrastive objectives (Konkle & Alvarez, 2022; Zhuang et al., 2021) produced representations closely aligned with activity in the human visual cortex, offering insight into the kinds of objective functions the brain may solve to construct visual representations. Extending this line of inquiry, Muttenthaler et al. (2022a) evaluated over 75 models spanning architectures, objectives, and datasets, and found that training data and objectives strongly predicted human-model alignment, whereas architecture and scale had minimal impact. This finding challenges scaling law accounts (Kaplan et al., 2020; Cherti et al., 2023) that predict generally larger models should always perform better (at most tasks). Other key findings regarding the importance of different computational ingredients include the finding that self-supervised contrastive methods align more closely with human vision than non-contrastive approaches (Chen et al., 2020; Zbontar et al., 2021; Caron et al., 2020); multimodal image–text contrastive training improves alignment over vision-only training (Radford et al., 2021; Jia et al., 2021; Pham et al., 2022); training on diverse, naturalistic datasets such as `JFT-3B` improves alignment beyond ImageNet (Zhai et al., 2022; Dehghani et al., 2023); enforcing shape bias on models enhances alignment while texture bias impairs it (Geirhos et al., 2019; Hermann et al., 2020); and intermediate layers often show stronger perceptual alignment than final layers (Berardino et al., 2017; Kumar et al., 2022). Similar approaches, dissecting the predictive power of different model building choices, have not been brought to bear in investigations into conceptual structure in LLMs- leaving unclear what the relative importance of different computational ingredients are when building foundation models of human semantic knowledge.

**Language Models and Conceptual Alignment** Recent work has begun extending analyses of representational alignment to language models (Suresh et al., 2023b; Marjieh et al., 2022). Early studies showed that contextualized embeddings from models like BERT and GPT-2 could predict human similarity judgments (Bommasani et al., 2020; Grand et al., 2022). More recent work demonstrates that LLMs can achieve remarkably high alignment with human conceptual structure (Marjieh et al., 2024a;b; Mukherjee et al., 2024; Suresh et al., 2025; 2023a; Mukherjee et al., 2023b), with instruction-tuned models showing particular advantages (Tong et al., 2024; Gurnee & Tegmark, 2024). Several studies have used triplet tasks specifically to probe LLM representations (Nam et al., 2024; Studdiford et al., 2025; Rathi et al., 2024; Suresh et al., 2023b), finding that larger models and those trained on more diverse data show better alignment. However, systematic analyses disentangling the effects of scale, architecture, training data, and objectives remain limited. Work on multimodal models suggests that vision-language training improves conceptual alignment beyond either modality alone (Yuksekgonul et al., 2023; Thrush et al., 2022; Qin et al., 2025), though the mechanisms remain unclear.

**Summary of key contributions.** We leverage the open-weight model ecosystem to construct a suite of 75+ models spanning diverse computational ingredients. Using concepts and semantic embeddings from THINGS dataset (Hebart et al., 2019) in conjunction with a triadic similarity judgment paradigm (Jamieson et al., 2015; Hebart et al., 2023; Sievert et al., 2023), we obtain both human and model-specific conceptual embeddings using analogous methods to ensure model-fair comparisons (Firestone, 2020). We then compare these embeddings to human data and apply statistical models to identify the ingredients most predictive of human-model alignment. Lastly, we

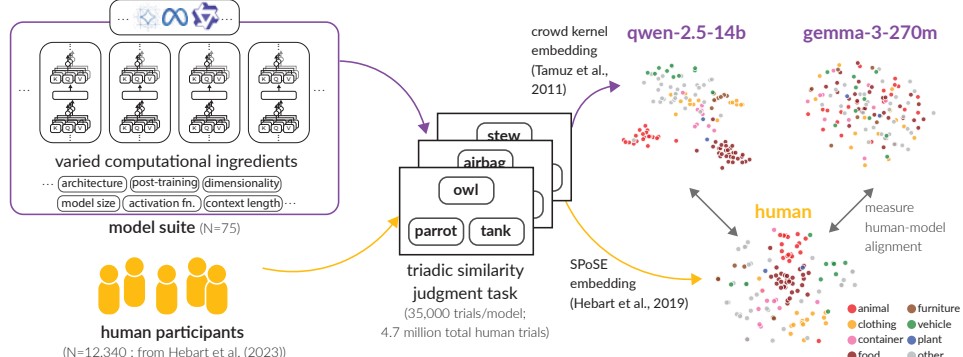

Figure 1: *Method for estimating human-model alignment.* We collected 35k triplet similarity judgments from each of the 77 models in our suite. We computed semantic embeddings based on these judgments and compared the representational geometry of model embeddings to human embeddings derived from Hebart et al. (2023).

relate alignment to performance on standard LLM benchmarks, highlighting points of convergence and divergence between representational alignment and task capabilities.

## 3 METHODS

**Concept Dataset**   To investigate how various LLM characteristics influence human alignment, we carefully curated a set of test concepts that would provide robust and interpretable results. Starting from the full set of 1,854 THINGS object concepts (Hebart et al., 2019; 2023), we subsampled 128 concrete, real-world object concepts (e.g., "lion", "banjo", "car") that spanned four main cognitive axes of variation: familiarity, artificiality, animacy, and size based on prior work (Mukherjee et al., 2023a). This sampling strategy ensured that we had adequate semantic diversity in our concept dataset (representative of the full THINGS database) while also ensuring that we could generate semantic embeddings for a large suite of models at scale in a tractable manner. The complete list of object concepts, along with detailed descriptions of our concept selection criteria and validation procedures, can be found in Appendix A.3.

**Model Suite**   We evaluate a range of open-source models varying across several key computational ingredients: model architecture, primary activation functions used, pre-training modality, whether models were instruction fine-tuned or not, scale (both in terms of model parameters and training tokens), context length, dimensionality of attention heads/MLPs/residual streams. While in principle it is possible to offer even more granular analyses of computational ingredients, these factors constitute major architectural and training decisions when building foundation models, which we believe provide adequate resolution to investigate the drivers of human-model alignment.

We evaluated a set of 77 open-weight transformer models including models from popular families (Llama, Qwen, Gemma, etc.). The full set of models can be seen in Table 4. Notably, we included models that varied in size from 100M parameters to 42B parameters, ranged in the degree of post-training (from base to instruction fine-tuned), and architecture-style (decoder-only, encoder-decoder, and state space models (SSMs) (Voelker & Eliasmith, 2018)).

**Triadic similarity judgment task**   The triadic similarity judgment task can be framed in two equivalent ways: (1) **anchored similarity judgments** where participants are presented with a triplet of items (images or words depicting different concepts) and select which of two option items is most similar to an anchored reference item, or **odd-one-out judgments** where they identify which of the three presented item is the odd-one-out. Both framings have been successfully employed to study conceptual representations in humans and models (Hebart et al., 2019; 2023; Mukherjee et al., 2023c; Colon & Rogers, 2023; Suresh et al., 2023b; Studdiford et al., 2025; Mukherjee et al.; Muttenthaler et al., 2022b). In this work, we leverage existing human similarity ratings data (Hebart et al., 2023) and corresponding embeddings, which were collected using the odd-one-out paradigm and embedded using the sparse positive similarity embedding (SPoSE) method, respectively. For model evaluations, we adopted the anchored similarity judgment paradigmn for computational efficiency, deriving embeddings using ordinal embedding techniques (Jamieson et al.,

2015; Tamuz et al., 2011) following recent work (Suresh et al., 2023b; Mukherjee et al.; Colon & Rogers, 2023). Crucially, the ordinal embedding approach provides theoretical bounds on sample complexity (Jamieson et al., 2015), allowing us to determine the number of triplet comparisons needed for faithful representations a priori (see Section 3 for details). While these approaches differ in framing and embedding computation, they yield comparable representational structures. This methodological choice allows us to maximize the use of existing high-quality, validated human data while efficiently collecting model responses with known sampling requirements in a scalable manner, ensuring robust human-model comparisons across a large suite of models.

**Human Semantic Embeddings**  Human semantic embeddings were obtained by Hebart et al. (2023) from the behavioral judgments of $N = 12,340$ online participants in a triplet odd-one-out task. On each trial, participants were shown three items from the THINGS database and were prompted to select the most dissimilar object(*"Which is the odd one out?"*). In total, 4.7 million behavioral triplet judgments were obtained for the full set of 1,854 concepts. To estimate an embedding space from the set of individual triplet judgments, Hebart et al. (2023) used the SPoSE algorithm (Hebart et al., 2023), which learns low-dimensional vectors for each object such that (1) pairs of objects judged as similar are placed closer in the embedding space, and (2) dimensions are constrained to be sparse in order to yield human-interpretable properties. The resulting space captured meaningful representational structure from the individual judgments of participants, revealing core dimensions along which human semantic representations are organized.

**Model Semantic Embeddings**  While much work assessing representational similarity between models and humans focus on extracting activation vectors from models and computing their similarity to human behavior-based representations (using methods like RSA (Kriegeskorte et al., 2008)), we instead estimated model semantic embeddings from model similarity judgments using a triadic comparison task akin to those often used to estimate human embeddings. We did so for two principled reasons. Firstly, while computations in transformer layers and attention heads have been linked to human language processing (Tuckute et al., 2024), there is no general framework for assessing which parts (layers, heads) of models (across different model families) are likely to carry signature of human semantic knowledge, making it difficult to assess what constitutes a fair comparison between model activations and human representations. Second, due to the ability of modern LLMs to respond to structured tasks when prompted in natural language and given the relative simplicity of the triadic judgment task (often single token responses suffice), we felt that administering the same test to both models and humans constitutes a species-fair (Firestone, 2020) comparison approach. Further, this allows for a unified embedding method to be applied to both systems ensuring that any discrepancies in alignment are not attributable to vastly different embedding methods.

For each model in our suite, we collected 35,000 triadic judgment responses. Using the human semantic embeddings, we estimated that a rank 30 space explains over 95% of the variance in embeddings (see A.7). Based on known bounds (Sievert et al., 2023), we require $(n \cdot d \cdot log_2(n)$[1]$)$ $\approx 26,900$ judgments on random sets of triplets to estimate a robust embedding. We collected 35,000 to account for noise in the judgments. Each triplet trial was randomly sampled from the $\binom{128}{3}$ possible triplet trials. We evaluated each model using its default huggingface sampling settings using a common system-level prompt — `"You are a helpful assistant who gives responses to questions"`. We used the following prompt for each trial for each model — `QUESTION: Which item is most similar to item_x: item_y or item_z? Respond only with the item name`. If we were evaluating a base model (not instruction-tuned), we appended `Answer:` after the initial prompt. Specific instruction template formats were applied where appropriate. Each prompt was evaluated independently.

Using these similarity judgments, we fit an ordinal embedding algorithm (Tamuz et al., 2011; Sievert et al., 2023) that organized the semantic embedding space such that items that were judged to be similar more often were pulled closer together in this space. We fit 30D embeddings based on the dimensionality of human semantic embeddings. We also held out 20% of the data during the embedding fitting process and evaluated the quality of the embeddings by monitoring the crowd-kernel loss (Tamuz et al., 2011) to ensure the embeddings were of high quality.

---

[1]$n$ is the number of items and $d$ is the expected dimensionality of the space

## 4 RESULTS AND DISCUSSION

We first report on how predictive different computational ingredients were of human-model alignment, initially considering each ingredient separately via individual correlations or $t$-tests (Kim, 2015), then assessing relative contributions of each via mixed linear models (Lindstrom & Bates, 1988). Finally, we evaluate the relationship between human-model alignment and model performance on other key LLM benchmarks (Han et al., 2024; Rein et al., 2023; Srivastava et al., 2023; Zhou et al., 2024; Hendrycks et al., 2020). Our primary alignment measure was the variance in human semantic embeddings explained by model embeddings ($R^2$) after Procrustes alignment (Gower, 1975). Procrustes transforms find the optimal linear transformation (rotation, scaling, translation) between two vector spaces of equal dimensionality to minimize squared distances between corresponding points. Using human embedding variance (human SS) as the baseline, we compute $R^2 = 1 - \frac{\text{residual SSE}}{\text{human SSE}}$. Other alignment measures such as CKA (Kornblith et al., 2019) and RSA(Kriegeskorte et al., 2008) were highly correlated with Procrustes $R^2$ (Appendix A.7).

### 4.1 CONTRIBUTIONS OF MODEL COMPUTATIONAL INGREDIENTS TO ALIGNMENT

**Instruction tuning leads to greater alignment.** Models with instruction fine-tuning were significantly better aligned than those without ($t = 5.11, p < 0.001$; see Figure 2A). Qualitatively, instruction fine-tuning clusters the representations of semantic categories, such that within-category items become more proximal in representation space and between-category items become more distal (see Appendix 7). Thus, the same post-training techniques that enable models to be more adept at following prompt instructions also bring their representations closer into alignment with humans'.

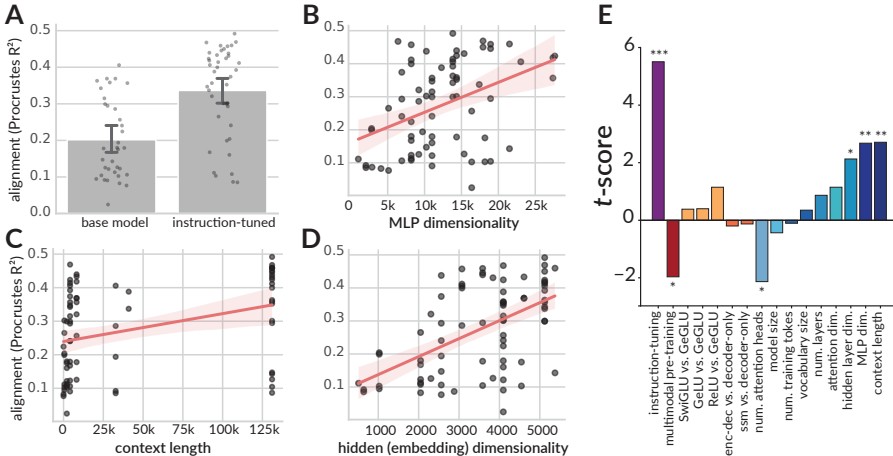

Figure 2: **Computational ingredients predictive of human model alignment**. **A.**, **B.**, **C.**, and **D.** show the relationship between instruction-tuning, MLP dimensionality, context length, and embedding dimensionality on Procrustes $R^2$. These were the four ingredients most predictive in the mixed-effects regression model. **E.** $t$-scores for each predictor, which highlight the relative contribution of each ingredient towards alignment when considered in a single statistical model.

**Model architecture dimensionality corresponds to greater alignment.** We found a strong positive relationship between the dimensionality of individual model architecture components — MLPs and attention heads — and the degree of model-human alignment. Specifically, the number of per-layer attention heads ($r = 0.52, p < 0.001$), attention matrix dimensionality ($r = 0.30, p = 0.026$), embeddings dimensionality ($r = 0.54, p < 0.001$), and MLP dimensionality ($r = 0.40, p < 0.001$) each correlated positively with human representational alignment. Figure 2B and D show the individual correlations between MLP and embedding dimensionality and model-human Procrustes $R^2$ (the attention dimensionality was not significant in multiple regression models reported in subsequent sections). Thus the greater expressivity granted by larger latent activation spaces and more attention heads predicts greater human alignment.

**Multi-modal pretraining has no effect on alignment.** While much of human concept learning is inherently multimodal (Rogers & McClelland, 2004; Patterson et al., 2007), there is limited work

showing whether vision-language pretraining (the most prominent class of multimodal training) leads to more aligned semantic representations (cf. (Qin et al., 2025)). In our model suite, however, multimodal (image) pre-training had no independent effect on human-model alignment relative to text-only models ($t = 0.34, p > 0.05$; Appendix Figure A.8).

**Larger model sizes showed greater human alignment.** For visual representations, Muttenthaler et al. (2022a) found that larger models need not yield more human-like representations. Neural scaling laws (Kaplan et al., 2020), however, might predict that larger and deeper models should be more performant, and perhaps more aligned. In accordance with this prediction, human-model alignment scaled approximately linearly with both model parameter count and the number of model layers ($r_{param} = 0.45, r_{layers} = 0.41, p < 0.001$; Appendix Figure A.8). Thus, larger scale predicts greater human-model alignment.

**Alignment increased with amount of training data.** Models exposed to more tokens in pretraining generally exhibited more human-like representations ($r = 0.33, p < 0.01$; Appendix Figure A.8). Although model training data also likely varied in quality and composition–thus limiting strong conclusions about which properties matter most–our results indicate that the *amount* of training data can predict alignment.

**Alignment scales with context length.** The triadic similarity task does not require a long context length, but we nevertheless found that models with a greater maximum context-length also demonstrated greater human alignment ($r = 0.35, p < 0.01$; Figure 2). This finding could be linked to post-training regimes that unlock long-context reasoning, which consequently lead models to perform well on short-context tasks as well.

**Choice of activation function does not affect alignment.** Activation functions have seen iterative development over the years with recent variants (e.g., SwiGLU (Shazeer, 2020)) being particularly well-suited to managing gradient flows during training and finetuning experiments leading to faster and more stable training regimes. But do choices in activation functions ground out in alignment? We found no advantage for any particular function in increasing human alignment ($F(3, 64) = 1.36, p = .26$; Appendix Figure A.8).

**Larger vocabulary size does not increase alignment.** One might expect that larger model vocabularies unlock greater expressivity for models, leading them to be more human-aligned. Against this prediction, we found no relationship between model vocabulary size and human-alignment ($r = 0.10, p > 0.05$; Appendix Figure A.8).

**Which computational ingredients have the greatest relative contribution towards human-model alignment?** Thus far, we have investigated how different computational ingredients impact human-model alignment in isolation. In reality, many of these ingredients vary across models at the same time, leaving open the questions of (1) which ingredients are most important *relative to others* and (2) which ingredients account for unique variance after others are considered.

To answer these questions, we fit a mixed-effects multiple regression model (Lindstrom & Bates, 1990) predicting Procrustes $R^2$ from *all* computational ingredients. Mixed-effects models combine the interpretability of OLS with random effects to capture item-level variance (Barr et al., 2013). We included random intercepts for model family (e.g., Llama, Qwen) to capture variance related to these families not grounded out in our set of ingredients, and fixed effects for all ingredients. The fitted model explained substantial variance in alignment ($R^2 = 0.78$; Fig. 2E). Instruction fine-tuning emerged as the strongest predictor ($\beta = 0.13$, $SE = 0.024$, $p < 0.001$). Dimensionality measures — MLP ($\beta = 0.049$, $SE = 0.018$, $p < 0.01$), embedding/hidden layers ($\beta = 0.048$, $SE = 0.022$, $p < 0.05$) — and context length ($\beta = 0.042$, $SE = 0.015$, $p < 0.01$) also accounted for significant unique variance. In contrast to the independent analyses, attention dimensionality was not significant, and more attention heads predicted worse alignment after accounting for other factors ($\beta = -0.045$, $p < 0.05$). Likewise, multimodal pretraining, which showed no independent effect on alignment, predicted reliably lower alignment after accounting for other factors ($\beta = -0.102$, $SE = 0.052$, $p < 0.05$). Thus, current multimodal training regimes may actually limit rather than improving human-model alignment. No other ingredient explained significant unique variance when others were included in the mode (see effect sizes in Figure 2E).

**A case study on the effects of post-training paradigms on alignment.** Modern LLM systems undergo various stages of post-training including *supervised fine-tuning* (SFT) (Ouyang et al., 2022),

*direct preference optimization* DPO (Rafailov et al., 2023), *reinforcement learning from human feedback* (RLHF) (Ouyang et al., 2022), and *instruction tuning* (IT) (Wei et al., 2021). It remains unclear what impact these post-training steps have on human-model alignment. Progress on this front has been limited due to a lack of open-weight models that have been checkpointed at the various possible stages. `OLMo` (Groeneveld et al., 2024) is a notable exception since it is available at four stages of post-training, including the base model (no post-training), SFT, DPO, and IT and is available at two model sizes (7B and 13B). `Instella` (Liu et al., 2025) and `Llama-3.1` (Meta AI, 2024) also have similar checkpointed model variants available with Instella having an additional stage of pretraining checkpointed.

Using these models, we asked whether successive stages of post-training led to models' representations becoming more aligned with humans'. Figure 3 shows how alignment changes as a function of post-training. Generally, we found that the SFT and DPO stages led to improvements in alignment whereas the effect of RLVR was more muted.

## 4.2 RELATIONSHIP BETWEEN ALIGNMENT AND ESTABLISHED BENCHMARKS

Prior work emphasizes that performance on a given model evaluation should "*translate to similar improvements on any other valid and reasonable evaluation data*" (Bowman & Dahl, 2021). But these arguments were developed in the context of specific NLP tasks, leaving it unclear whether this notion extends to representational alignment. To understand the relationship between alignment and performance on standard LLM benchmarks we estimated the correlation between alignment, measured using three metrics (Procrustes $R^2$, CKA, and RSA correlation), and model scores on five key benchmarks – `MMLU`, `IFEval`, `BigBenchHard` (`BBH`), `GPQA`, and `MUSR`. Figure 4A shows the relationship between alignment (Procrustes $R^2$) and scores on BBH, the benchmark most strongly correlated with alignment.

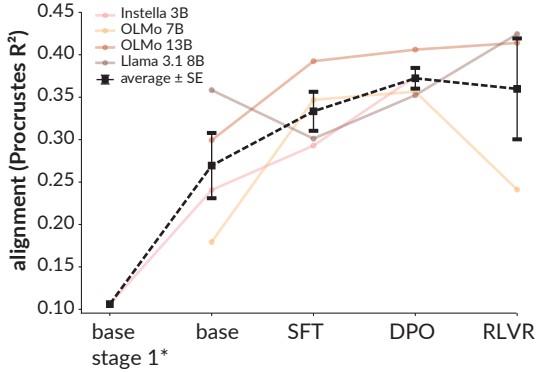

While models that performed well on BBH generally had stronger alignment, we found some notable divergences especially for base models that scored lower on alignment than one might expect based on their BBH score. Figure 4B shows the correlations between alignment and all the established benchmarks. While all three alignment metrics were in agreement, we found that the overall correlations varied ($r_{min} = 0.36, p < 0.05$; $r_{max} = 0.74, p < 0.001$). We focus on Procrustes $R^2$ moving forward.

Performance on benchmarks that probe probe domain-general knowledge (BBH; $r = 0.74, p < 0.001$; MMLU; $r = 0.71, p < 0.001$) and instruction-following ability (IFEval; $r = 0.68, p < 0.001$) were

Figure 3: **Effects of post-training on alignment**. The black points show mean alignment (across models) at each post-training stage.*Instella only.

more correlated with alignment than benchmarks testing more specific domain knowledge and reasoning ability (Math; $r = 0.61, p < 0.001$, MUSR; $r = 0.36, p < 0.05$).

This finding is sensible given that alignment with human semantic judgments requires representing semantic *knowledge* broadly in human-like ways (captured by `BBH` and `MMLU`) and *deploying* that knowledge in context-sensitive ways according to instructions (captured by `IFEval`). We also found that the correlation between `BBH` and `IFEval` was significantly weaker than the correlation between these evaluations and our measure of representational alignment($r_{BBH,IFEval} = 0.44, p < 0.01$), suggesting our alignment measure captures separate competencies unique to each benchmark.

## 5 CONCLUSION

The rapid advancement of language models (both in scale and diversity) and the varied, often unpredictable, performance of models on standard benchmarks has left open the question "what

does it take to make human-like models"? A gold standard for 'human-like' would be to have models that not only perform tasks in human-like ways but also have internal representations and computations that are similar to that of humans. Here we attempt to identify the computational ingredients most predictive of human-model representational alignment.

We first leveraged a large-scale dataset of human triadic similarity judgments and associated semantic embeddings from the THINGS dataset and set them as the target for measuring alignment. Next, we carefully constructed a suite of 77 open-weight language models that varied in several key computational ingredients including model scale, training dataset size, dimensionality of model embeddings, and post-training regimes, among others. We estimated semantic embeddings for each model using a triadic similarity judgment task analogous to the human task and measured representational alignment between model and human embeddings as function of different model characteristics. We arrived at several key insights.

First, we found two convergent pieces of evidence that modern post-training techniques are central to human-model alignment: (1) instruction-finetuning was the strongest predictor of alignment even after controlling for variance explained by other ingredients (Figure 2 and (2) alignment tended to increase across different stages of post-training (Figure 3). The importance of context length in predicting alignment further suggests that the methods that enable long context-reasoning also bring model representations into closer alignment with humans. Second, we observed that architectural dimensionality (embedding layers, MLP) were relatively more important than overall model size and training corpus size in predicting alignment, especially in the mixed-effects model suggesting that representational capacity is key to developing models that align with human representations. Third, we found that while some existing benchmarks were correlated with our measure of alignment (e.g., *MMLU, BBH*), no single benchmark captured all the variance in representational alignment. Notably, benchmarks emphasizing broad semantic knowledge and its flexible deployment showed the strongest correlations, consistent with theories of how humans represent semantic knowledge (Rogers & McClelland, 2004; Saxe et al., 2019).

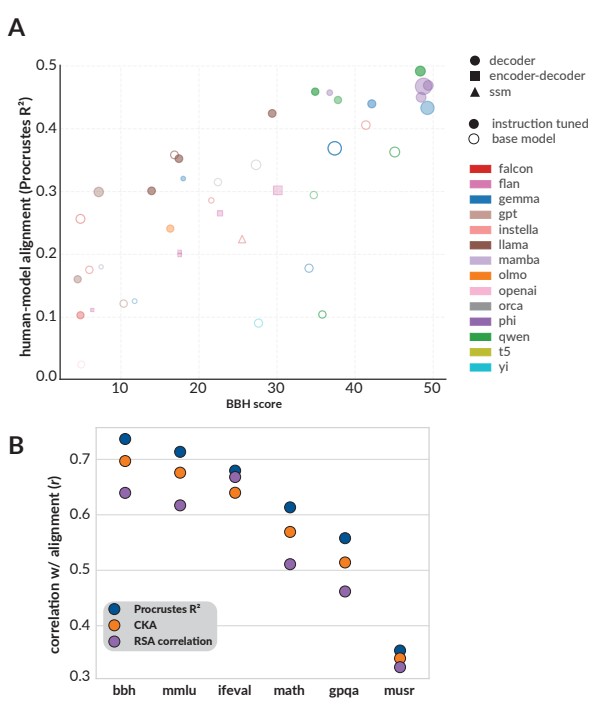

Figure 4: **Relationship between alignment and other LLM benchmarks**. **A.** `BigBenchHard` scores for each model in our suite vs. their Procrustes $R^2$ value w.r.t. human embeddings. **B.** Correlation between alignment and the six LLM benchmarks evaluated.

**Limitations** To our knowledge, this is the first attempt to isolate the computational ingredients predictive of human–LLM conceptual alignment in language models, however several limitations remain that future work could address. First, while we evaluated many core computational ingredients, we could not assess the role of training dataset composition (e.g., text–code balance, language coverage) due to limited documentation for many models. Future work should look beyond token counts to clarify which dataset properties enable semantic alignment. Second, some classes of models were sparsely represented in our suite including SSMs, multimodal models, and mixture-of-expert models limiting the reliability of insights we can derive regarding ingredients that support them. Third, while similarity judgements are a well validated method and scale well to large concept sets, there may be other ways of probing semantic knowledge such as by observing how models deploy this knowledge in open-ended reasoning tasks (Wong et al., 2025; Mahowald et al., 2024). Future work can seek to study such tasks that tackle the *deployment* of semantic knowledge.

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

# A APPENDIX

## A.1 ODD-ONE-OUT JUDGMENTS → SPoSE ESTIMATION (HUMAN SEMANTIC EMBEDDINGS)

To establish a ground-truth human semantic space, we employ the SPoSE (Sparse Positive Similarity Embedding) framework applied to odd-one-out behavioral data from the THINGS dataset (Hebart et al., 2023).

*SPoSE Algorithm.* Let $\mathcal{X} = \{1, \ldots, n\}$ index $n$ items and let $X \in \mathbb{R}_{\geq 0}^{n \times d}$ be a nonnegative, low-dimensional embedding with rows $x_i^\top \in \mathbb{R}_{\geq 0}^d$. We define pairwise similarities using the inner product $s_{ij} = x_i^\top x_j$, yielding similarity matrix $S = X X^\top$. In an odd-one-out trial with unordered triplet $\{i, j, k\}$, participants select the *most similar pair* among $\{(i, j), (i, k), (j, k)\}$. SPoSE models this choice behavior with a multinomial logit over pairwise similarities:

$$\Pr\big[(i, j) \text{ chosen} \mid \{i, j, k\}, X\big] = \frac{\exp(s_{ij})}{\exp(s_{ij}) + \exp(s_{ik}) + \exp(s_{jk})}. \tag{1}$$

*Estimation and Implementation.* The SPoSE estimate $\hat{X}$ is obtained via a sparsity-promoting MAP program that encourages both nonnegativity and interpretability:

$$\hat{X} \in \arg\max_{X \geq 0} \Big\{ \underbrace{\sum_{t=1}^{T} \log p\big(y_t \mid \{i_t, j_t, k_t\}, X\big)}_{\text{log-likelihood}} - \lambda \|X\|_1 \Big\}, \tag{2}$$

where $y_t$ is the chosen pair on trial $t$, and $\lambda > 0$ controls sparsity.[2] We use the publicly released SPoSE embeddings as our human reference space.

*Dimensionality Selection.* To determine the appropriate embedding dimension, we apply PCA to the released SPoSE matrix and retain the smallest $d$ that explains at least $95\%$ of the variance (Appendix A.7). For our corpus of $n = 128$ concepts, this yields $d = \hat{d}$ dimensions (*[fill in]*), which we use as the target dimensionality for model embeddings.

## A.2 ANCHORED SIMILARITY JUDGMENTS → ORDINAL EMBEDDING (MODEL SEMANTIC EMBEDDINGS)

To obtain comparable model-based semantic embeddings, we apply an ordinal embedding algorithm (Tamuz et al., 2011; Sievert et al., 2023) to model similarity judgments, creating semantic spaces where frequently co-judged similar items are positioned closer together.

*Embedding Algorithm.* Each triplet $\{i, j, k\}$ from model judgments yields an ordinal constraint: "$i$ is closer to $j$ than to $k$." Let $\{x_i^\star\}_{i=1}^n \subset \mathbb{R}^d$ denote the unknown true point locations we seek to estimate, with corresponding (squared) Euclidean distance matrix $D^\star$. We model the probability of observing each ordinal constraint using a standard noisy triplet model with monotone link function:

$$\Pr\big[y_{(i;j,k)} = 1\big] = f\big(D_{ik}^\star - D_{ij}^\star\big), \qquad f(0) = \tfrac{1}{2}. \tag{3}$$

*Estimation and Implementation.* The ordinal embedding algorithm minimizes crowd-kernel triplet loss (Tamuz et al., 2011) by optimizing Euclidean distances between item pairs. We minimize an empirical surrogate of the negative log-likelihood using a low-rank Gram matrix parameterization. To ensure reliable estimates, we reserve 20% of our triplet data for validation purposes and monitor crowd-kernel loss throughout the fitting procedure.

*Sample Complexity Considerations.* When the true embedding has rank $d$ and triplet judgments are sampled approximately uniformly at random, finite-sample theory provides the out-of-sample prediction error bound:

$$\mathcal{E}_{\text{pred}} = \tilde{O}\bigg(\sqrt{\frac{d\,n \log n}{|S|}}\bigg), \qquad \Rightarrow \quad |S| = \tilde{\Theta}(d\,n \log n). \tag{4}$$

---

[2]The nonnegativity constraint and elementwise $\ell_1$ penalty are equivalent to an exponential prior on factors and empirically yield interpretable semantic dimensions.

Thus, $\tilde{\Theta}(nd \log n)$ triplet judgments suffice for accurate prediction of new comparisons, and at least $\Omega(d\, n \log n)$ ordinal comparisons are information-theoretically necessary (Jain et al., 2016). Guided by this theory and setting the model embedding dimension to match the human-derived $\hat{d}$, we collect

$$N_{\text{triplets}} \;=\; c\, d\, n \log n \tag{5}$$

triplet judgments per model, where $c$ is chosen to balance statistical efficiency with computational constraints.

This parallel approach provides a principled framework for human-model semantic comparison. Both methods yield $d$-dimensional embeddings where geometric relationships reflect semantic similarities, with the SPoSE framework extracting interpretable human semantic structure and ordinal embedding converting model judgments into geometrically comparable representations with theoretically grounded sample complexity $\tilde{\Theta}(d\, n \log n)$.

Table 1: Model Alignment Across Training Stages

| Model | Training Stage | Alignment (R²) |
|---|---|---|
| Instella 3B | Base Stage 1 | 0.106 |
| Instella 3B | Base | 0.241 |
| Instella 3B | SFT | 0.293 |
| Instella 3B | DPO | 0.374 |
| Llama 3.1 8B | Base | 0.358 |
| Llama 3.1 8B | SFT | 0.301 |
| Llama 3.1 8B | DPO | 0.352 |
| Llama 3.1 8B | RLVR | 0.424 |
| OLMo 13B | Base | 0.299 |
| OLMo 13B | SFT | 0.392 |
| OLMo 13B | DPO | 0.406 |
| OLMo 13B | RLVR | 0.414 |
| OLMo 7B | Base | 0.179 |
| OLMo 7B | SFT | 0.347 |
| OLMo 7B | DPO | 0.357 |
| OLMo 7B | RLVR | 0.241 |
| Average | Base Stage 1 | 0.106 |
| Average | Base | 0.269 |
| Average | SFT | 0.333 |
| Average | DPO | 0.372 |
| Average | RLVR | 0.360 |

## A.3 OBJECT CONCEPTS

The following table presents the 128 concrete object concepts from the THINGS dataset used in our experiments, along with their semantic categories and definitions.

Table 2: Complete list of object concepts with categories and definitions

| Concept | Category | Definition |
|---|---|---|
| **Animal** | | |
| badger | animal | sturdy carnivorous burrowing mammal with strong claws; widely distributed in the northern hemisphere |
| bear | animal | massive plantigrade carnivorous or omnivorous mammals with long shaggy coats and strong claws |
| butterfly | animal | diurnal insect typically having a slender body with knobbed antennae and broad colorful wings |

**Table 2 continued from previous page**

| Concept | Category | Definition |
|---|---|---|
| chipmunk | animal | a burrowing ground squirrel of western America and Asia; has cheek pouches and a light and dark stripe running down the body |
| cow | animal | female of domestic cattle: "'moo-cow' is a child's term" |
| crow | animal | black birds having a raucous call |
| fly | animal | two-winged insects characterized by active flight |
| giraffe | animal | tallest living quadruped; having a spotted coat and small horns and very long neck and legs; of savannahs of tropical Africa |
| grasshopper | animal | terrestrial plant-eating insect with hind legs adapted for leaping |
| hyena | animal | doglike nocturnal mammal of Africa and southern Asia that feeds chiefly on carrion |
| iguana | animal | large herbivorous tropical American arboreal lizards with a spiny crest along the back; used as human food in Central America and South America |
| lion | animal | large gregarious predatory feline of Africa and India having a tawny coat with a shaggy mane in the male |
| mosquito | animal | two-winged insect whose female has a long proboscis to pierce the skin and suck the blood of humans and animals |
| mouse | animal | a hand-operated electronic device that controls the coordinates of a cursor on your computer screen as you move it around on a pad; on the bottom of the device is a ball that rolls on the surface of the pad |
| owl | animal | nocturnal bird of prey with hawk-like beak and claws and large head with front-facing eyes |
| parrot | animal | usually brightly colored zygodactyl tropical birds with short hooked beaks and the ability to mimic sounds |
| puffin | animal | any of two genera of northern seabirds having short necks and brightly colored compressed bills |
| snail | animal | freshwater or marine or terrestrial gastropod mollusk usually having an external enclosing spiral shell |
| snake | animal | limbless scaly elongate reptile; some are venomous |
| tarantula | animal | large hairy tropical spider with fangs that can inflict painful but not highly venomous bites |

**Clothing**

| Concept | Category | Definition |
|---|---|---|
| bathrobe | clothing | a loose-fitting robe of towelling; worn after a bath or swim |
| beanie | clothing | a small skullcap; formerly worn by schoolboys and college freshmen |
| bow | clothing | a knot with two loops and loose ends; used to tie shoelaces |
| bracelet | clothing | jewelry worn around the wrist for decoration |
| button | clothing | a round fastener sewn to shirts and coats etc to fit through buttonholes |
| chaps | clothing | (usually in the plural) leather leggings without a seat; joined by a belt; often have flared outer flaps; worn over trousers by cowboys to protect their legs |
| hat | clothing | headdress that protects the head from bad weather; has shaped crown and usually a brim |
| jeans | clothing | (usually plural) close-fitting trousers of heavy denim for manual work or casual wear |
| kilt | clothing | a knee-length pleated tartan skirt worn by men as part of the traditional dress in the Highlands of northern Scotland |
| kimono | clothing | a loose robe; imitated from robes originally worn by Japanese |
| kneepad | clothing | protective garment consisting of a pad worn by football or baseball or hockey players |
| tuxedo | clothing | semiformal evening dress for men |
| uniform | clothing | clothing of distinctive design worn by members of a particular group as a means of identification |
| veil | clothing | a garment that covers the head and face |
| visor | clothing | a brim that projects to the front to shade the eyes |

**Table 2 continued from previous page**

| Concept | Category | Definition |
| --- | --- | --- |
| **Container** | | |
| bag | container | a flexible container with a single opening |
| cooker | container | a utensil for cooking |
| doily | container | a small round piece of linen placed under a dish or bowl |
| fishbowl | container | a transparent bowl in which small fish are kept |
| honeypot | container | South African shrub whose flowers when open are cup-shaped resembling artichokes |
| thermos | container | vacuum flask that preserves temperature of hot or cold drinks |
| vase | container | an open jar of glass or porcelain used as an ornament or to hold flowers |
| **Food** | | |
| appetizer | food | food or drink to stimulate the appetite (usually served before a meal or as the first course) |
| applesauce | food | puree of stewed apples usually sweetened and spiced |
| baklava | food | rich Middle Eastern cake made of thin layers of flaky pastry filled with nuts and honey |
| beer | food | a general name for alcoholic beverages made by fermenting a cereal (or mixture of cereals) flavored with hops |
| bok choy | food | Asiatic plant grown for its cluster of edible white stalks with dark green leaves |
| bread | food | food made from dough of flour or meal and usually raised with yeast or baking powder and then baked |
| crepe | food | small very thin pancake |
| cupcake | food | small cake baked in a muffin tin |
| dessert | food | a dish served as the last course of a meal |
| dough | food | a flour mixture stiff enough to knead or roll |
| enchilada | food | tortilla with meat filling baked in tomato sauce seasoned with chili |
| grapefruit | food | large yellow fruit with somewhat acid juicy pulp; usual serving consists of a half |
| gravy | food | a sauce made by adding stock, flour, or other ingredients to the juice and fat that drips from cooking meats |
| hummus | food | a thick spread made from mashed chickpeas, tahini, lemon juice and garlic; used especially as a dip for pita; originated in the Middle East |
| leek | food | plant having a large slender white bulb and flat overlapping dark green leaves; used in cooking; believed derived from the wild Allium ampeloprasum |
| mango | food | large oval tropical fruit having smooth skin, juicy aromatic pulp, and a large hairy seed |
| margarita | food | a cocktail made of tequila and triple sec with lime and lemon juice |
| mashed potato | food | potato that has been peeled and boiled and then mashed |
| milkshake | food | frothy drink of milk and flavoring and sometimes fruit or ice cream |
| oatmeal | food | porridge made of rolled oats |
| parfait | food | layers of ice cream and syrup and whipped cream |
| pumpkin | food | usually large pulpy deep-yellow round fruit of the squash family maturing in late summer or early autumn |
| quesadilla | food | a tortilla that is filled with cheese and heated |
| quiche | food | a tart filled with rich unsweetened custard; often contains other ingredients (as cheese or ham or seafood or vegetables) |
| ravioli | food | small circular or square cases of dough with savory fillings |
| sea urchin | food | shallow-water echinoderms having soft bodies enclosed in thin spiny globular shells |
| souffle | food | light fluffy dish of egg yolks and stiffly beaten egg whites mixed with e.g. cheese or fish or fruit |
| stew | food | food prepared by stewing especially meat or fish with vegetables |

**Table 2 continued from previous page**

| Concept | Category | Definition |
|---|---|---|
| tortellini | food | small ring-shaped stuffed pasta |
| waffle | food | pancake batter baked in a waffle iron |
| wrap | food | a sandwich in which the filling is rolled up in a soft tortilla |

**Furniture**

| Concept | Category | Definition |
|---|---|---|
| bassinet | furniture | a basket (usually hooded) used as a baby's bed |
| bathtub | furniture | a relatively large open container that you fill with water and use to wash the body |
| beanbag | furniture | a small cloth bag filled with dried beans; thrown in games |
| bed | furniture | a piece of furniture that provides a place to sleep |
| bench | furniture | a long seat for more than one person |
| coaster | furniture | a covering (plate or mat) that protects the surface of a table (i.e., from the condensation on a cold glass or bottle) |
| computer screen | furniture | a screen used to display the output of a computer to the user |
| cot | furniture | a small bed that folds up for storage or transport |
| couch | furniture | an upholstered seat for more than one person |
| crib | furniture | baby bed with high sides made of slats |

**Other**

| Concept | Category | Definition |
|---|---|---|
| album | other | a book of blank pages with pockets or envelopes; for organizing photographs or stamp collections etc |
| backscratcher | other | a long-handled scratcher for scratching your back |
| banjo | other | a stringed instrument of the guitar family that has long neck and circular body |
| baseball | other | a ball used in playing baseball |
| baseball glove | other | the handwear used by fielders in playing baseball |
| bassoon | other | a double-reed instrument; the tenor of the oboe family |
| beachball | other | large and light ball; for play at the seaside |
| blower | other | a device that produces a current of air |
| bongo | other | a small drum; played with the hands |
| cannonball | other | a solid projectile that in former times was fired from a cannon |
| canvas | other | an oil painting on canvas fabric |
| chainsaw | other | portable power saw; teeth linked to form an endless chain |
| cymbal | other | a percussion instrument consisting of a concave brass disk; makes a loud crashing sound when hit with a drumstick or when two are struck together |
| doorknob | other | a knob used to release the catch when opening a door (often called 'doorhandle' in Great Britain) |
| doorknocker | other | a device (usually metal and ornamental) attached by a hinge to a door |
| extinguisher | other | a manually operated device for extinguishing small fires |
| fire alarm | other | a device that makes a loud sound to warn people when there is a fire |
| guitar | other | a stringed instrument usually having six strings; played by strumming or plucking |
| hatchet | other | a small ax with a short handle used with one hand (usually to chop wood) |
| hula hoop | other | a large hoop spun around the body by gyrating the hips, for play or exercise. |
| knitting needle | other | needle consisting of a slender rod with pointed ends; usually used in pairs |
| knob | other | a round handle |
| lawnmower | other | garden tool for mowing grass on lawns |
| pinball | other | a game played on a sloping board; the object is to propel marbles against pins or into pockets |
| pocket watch | other | a watch that is carried in a small watch pocket |

**Table 2 continued from previous page**

| Concept | Category | Definition |
|---------|----------|------------|
| shuffleboard | other | a game in which players use long sticks to shove wooden disks onto the scoring area marked on a smooth surface |
| skeleton | other | the hard structure (bones and cartilages) that provides a frame for the body of an animal |
| swab | other | implement consisting of a small piece of cotton that is used to apply medication or cleanse a wound or obtain a specimen of a secretion |
| tennis ball | other | ball about the size of a fist used in playing tennis |
| toilet paper | other | a soft thin absorbent paper for use in toilets |
| treadmill | other | an exercise device consisting of an endless belt on which a person can walk or jog without changing place |
| trowel | other | a small hand tool with a handle and flat metal blade; used for scooping or spreading plaster or similar materials |
| volleyball | other | an inflated ball used in playing volleyball |
| **Plant** | | |
| flower | plant | a plant cultivated for its blooms or blossoms |
| gourd | plant | any of numerous inedible fruits with hard rinds |
| **Vehicle** | | |
| airbag | vehicle | a safety restraint in an automobile; the bag inflates on collision and prevents the driver or passenger from being thrown forward |
| carriage | vehicle | a vehicle with wheels drawn by one or more horses |
| exhaust pipe | vehicle | a pipe through which burned gases travel from the exhaust manifold to the muffler |
| hot-air balloon | vehicle | balloon for travel through the air in a basket suspended below a large bag of heated air |
| humvee | vehicle | a high mobility, multipurpose, military vehicle with four-wheel drive |
| missile | vehicle | a rocket carrying a warhead of conventional or nuclear explosives; may be ballistic or directed by remote control |
| odometer | vehicle | a meter that shows mileage traversed |
| taillight | vehicle | lamp (usually red) mounted at the rear of a motor vehicle |
| tank | vehicle | an enclosed armored military vehicle; has a cannon and moves on caterpillar treads |
| taxi | vehicle | a car driven by a person whose job is to take passengers where they want to go in exchange for money |

## A.4  MODEL SUITE

Table 3: List of all the models used

| Model | | Attention Type | Training Tokens | #Heads | Hidden Dim |
|-------|---|---------|---------|--------|-----------|
| **Falcon3-3B-Base** Team (2024) | Falcon-LLM | GQA | 100B | 12 (4 KV) | 3072 |
| **tiiuae/falcon-7b** Team (2024) | Falcon-LLM | MQA | 1.5T | 71 | 4544 |
| **tiiuae/falcon-7b-instruct** Falcon-LLM Team (2024) | | MQA | | 71 | 4544 |
| **Falcon3-10B-Base** Team (2024) | Falcon-LLM | GQA | 2T | 12 (4 KV) | 3072 |
| **tiiuae/falcon-11b** Team (2024) | Falcon-LLM | GQA | 5T | 32 (8 KV) | 4096 |

**Table 3 continued from previous page**

| Model | Attention Type | Training Tokens | #Heads | Hidden Dim |
|---|---|---|---|---|
| **flan-t5-small** Chung et al. (2022) | Multi-Head | | 8 | 512 |
| **flan-t5-large** Chung et al. (2022) | Multi-Head | | 16 | 1024 |
| **flan-t5-xl** Chung et al. (2022) | Multi-Head | | 32 | 1024 |
| **flan-t5-xxl** Chung et al. (2022) | Multi-Head | | 128 | 1024 |
| **gemma-3-270m** Ji et al. (2024) | GQA | | 8 (shared KV) | 2048 |
| **gemma-3-270m-it** Ji et al. (2024) | GQA | | 8 (shared KV) | 2048 |
| **gemma-2-2b** Ji et al. (2024) | MQA | | 8 (shared KV) | 2048 |
| **gemma-2-2b-it** Ji et al. (2024) | MQA | | 8 (shared KV) | 2048 |
| **gemma-2-9b** Ji et al. (2024) | MHA | | 16 | 4096 |
| **gemma-2-9b-it** Ji et al. (2024) | MHA | | 16 | 4096 |
| **gemma-2-27b** Ji et al. (2024) | MHA | | 16 | 6144 |
| **gemma-2-27b-it** Ji et al. (2024) | MHA | | 16 | 6144 |
| **gemma-3-4b-it** Ji et al. (2024) | GQA | | 12 (4 KV) | 3072 |
| **gemma-3-4b-pt** Ji et al. (2024) | GQA | | 12 (4 KV) | 3072 |
| **gemma-3-12b-it** Ji et al. (2024) | GQA | | 16 (4 KV) | 4096 |
| **gemma-3-12b-pt** Ji et al. (2024) | GQA | | 16 (4 KV) | 4096 |
| **gemma-3-27b** Ji et al. (2024) | GQA | | 16 (4 KV) | 5120 |
| **gemma-3-27b-it** Ji et al. (2024) | GQA | | 16 (4 KV) | 5120 |
| **GPT-OSS-20B** Black et al. (2022) | Multi-Head | ∼300B | 64 | 6144 |
| **amd/Instella-3B** Liu et al. (2025) | Multi-Head | 4.15T | 32 | 2560 |
| **amd/Instella-3B-Instruct** Liu et al. (2025) | Multi-Head | | 32 | 2560 |
| **Llama-3.1-8B-Instruct** Meta AI (2024) | MHA | | 32 | 4096 |
| **state-spaces/mamba-1.4b-hf** Dao et al. (2023) | (SSM) | | | |
| **allenai/OLMo-2-1124-7B** Groeneveld et al. (2024) | MHA | | 32 | 4096 |
| **allenai/OLMo-2-1124-7B-Instruct** Groeneveld et al. (2024) | MHA | | 32 | 4096 |
| **allenai/OLMo-2-1124-13B-Instruct** Groeneveld et al. (2024) | MHA | | 40 | 5120 |
| **OLMo-2-1124-7B-DPO** Groeneveld et al. (2024) | MHA | | 32 | 4096 |
| **OLMo-2-1124-7B-SFT** Groeneveld et al. (2024) | MHA | | 32 | 4096 |
| **AMD-OLMo-1B** Groeneveld et al. (2024) | MHA | | 16 | 2048 |
| **AMD-OLMo-1B-SFT-DPO** Groeneveld et al. (2024) | MHA | | 16 | 2048 |
| **GPT-J-6B** EleutherAI (2021) | Multi-Head | ∼300B | 16 | 4096 |
| **orca-2-7b** Mukherjee et al. (2023c) | Multi-Head | | 32 | 4096 |
| **orca-2-13b** Mukherjee et al. (2023c) | Multi-Head | | 40 | 5120 |
| **phi-3.5-mini-instruct** Gunasekar et al. (2023) | Multi-Head | | 32 | 2048 |
| **phi-3-mini-128k-instruct** Gunasekar et al. (2023) | Multi-Head | | 32 | 2048 |
| **phi-3-medium-128k-instruct** Gunasekar et al. (2023) | Multi-Head | | 40 | 5120 |
| **Phi-3-medium-4k-instruct** Gunasekar et al. (2023) | Multi-Head | | 40 | 5120 |

**Table 3 continued from previous page**

| Model | Attention Type | Training Tokens | #Heads | Hidden Dim |
|---|---|---|---|---|
| **phi-1_5** Gunasekar et al. (2023) | Multi-Head | 7B | 32 | 2048 |
| **qwen-14b** Qwen Team (2023) | GQA | 3T | 40 (10 KV) | 5120 |
| **qwen-14b-chat** Qwen Team (2023) | GQA | | 40 (10 KV) | 5120 |
| **qwen-2-7b** Qwen Team (2023) | GQA | | 32 (8 KV) | 4096 |
| **qwen-2-7b-instruct** Qwen Team (2023) | GQA | | 32 (8 KV) | 4096 |
| **qwen-2.5-7b** Qwen Team (2023) | GQA | | 32 (8 KV) | 4096 |
| **qwen-2.5-7b-instruct** Qwen Team (2023) | GQA | | 32 (8 KV) | 4096 |
| **qwen-2.5-14b** Qwen Team (2023) | GQA | | 40 (10 KV) | 5120 |
| **qwen-2.5-14b-instruct** Qwen Team (2023) | GQA | | 40 (10 KV) | 5120 |
| **t5-3b** Raffel et al. (2020) | Multi-Head | ∼1T | 32 | 1024 |
| **t5-11b** Raffel et al. (2020) | Multi-Head | ∼1T | 128 | 1024 |
| **yi-9B** 01.AI Team (2024) | Multi-Head | 3.1T | 32 | 6144 |
| **yi-9B (coder)** 01.AI Team (2024) | Multi-Head | 0.8T | 32 | 6144 |

## A.5 DETAILS FOR SPOSE (ODD-ONE-OUT) ESTIMATION

Given $S = XX^\top$ with $X \geq 0$, the triplet likelihood is the three-way softmax in equation 1. The MAP program equation 2 is optimized by minibatch stochastic gradients; $\lambda$ is tuned by held-out triplets. The released THINGS-data SPoSE embeddings are derived from large-scale odd-one-out judgments and approach the behavioral noise ceiling in predicting left-out triplets, despite far fewer than the $O(n^3)$ triplets required to fully determine a dense similarity matrix (**?**).

## A.6 ORDINAL EMBEDDING: RISK BOUNDS AND RECOVERY

Let $G^\star$ be the centered Gram matrix of $\{x_i^\star\} \subset \mathbb{R}^d$ and let $\mathcal{L}(G)$ denote the expected logistic triplet loss induced by equation 3. If $G^\star$ has rank $d$ and $|S|$ triplets are drawn uniformly at random, then with probability $1 - \delta$,

$$\mathcal{L}(\hat{G}) - \mathcal{L}(G^\star) \leq C_1 \sqrt{\frac{d \, n \log n}{|S|}} + C_2 \sqrt{\frac{\log(1/\delta)}{|S|}}, \tag{6}$$

for universal constants $C_1, C_2$ depending on the loss Lipschitz constant (**?**). Moreover, writing $C^\star$ for the component of the EDM orthogonal to the linear operator's kernel, one can recover the distance structure with

$$\frac{1}{n^2} \|\hat{C} - C^\star\|_F^2 = \tilde{O}\left(\frac{d \, n \log n}{|S|}\right), \tag{7}$$

which implies equation **??** samples suffice up to logarithmic factors; $\Omega(d \, n \log n)$ lower bounds are known (**?**).

## A.7 DIMENSIONALITY SELECTION FROM HUMAN SPOSE

We compute PCA on the released SPoSE embedding matrix (items $\times$ dimensions), retain the top $d$ components explaining at least $95\%$ variance, and set $d = \hat{d}$ for model-side ordinal embeddings. In our corpus ($n = 128$), this yields $\hat{d} = 29$, and therefore $N_{\text{triplets}} = c \, \hat{d} \, n \log n = 31,288$ triplets per model where $c = 4$. We increase $N_{\text{triplets}}$ to **35,000** to provide a margin of error.

## A.8 MEASURE CORRELATIONS

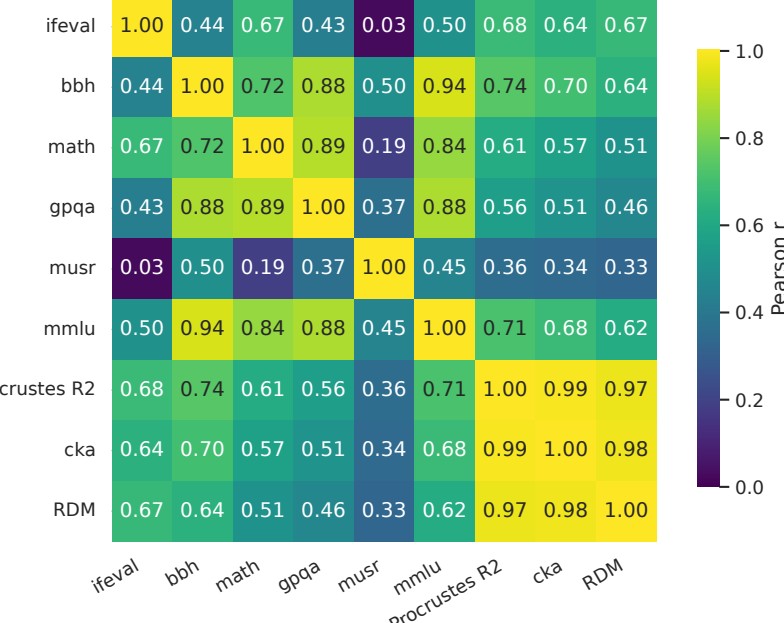

Figure 5: Correlations between measures of human-model alignment and model benchmarks, for models where benchmark scores are publicly are available. (Procrustes $R^2$, CKA, RSM)

## A.9 RELATIONSHIPS BETWEEN COMPUTATIONAL INGREDIENT PROPERTIES AND HUMAN-MODEL ALIGNMENT

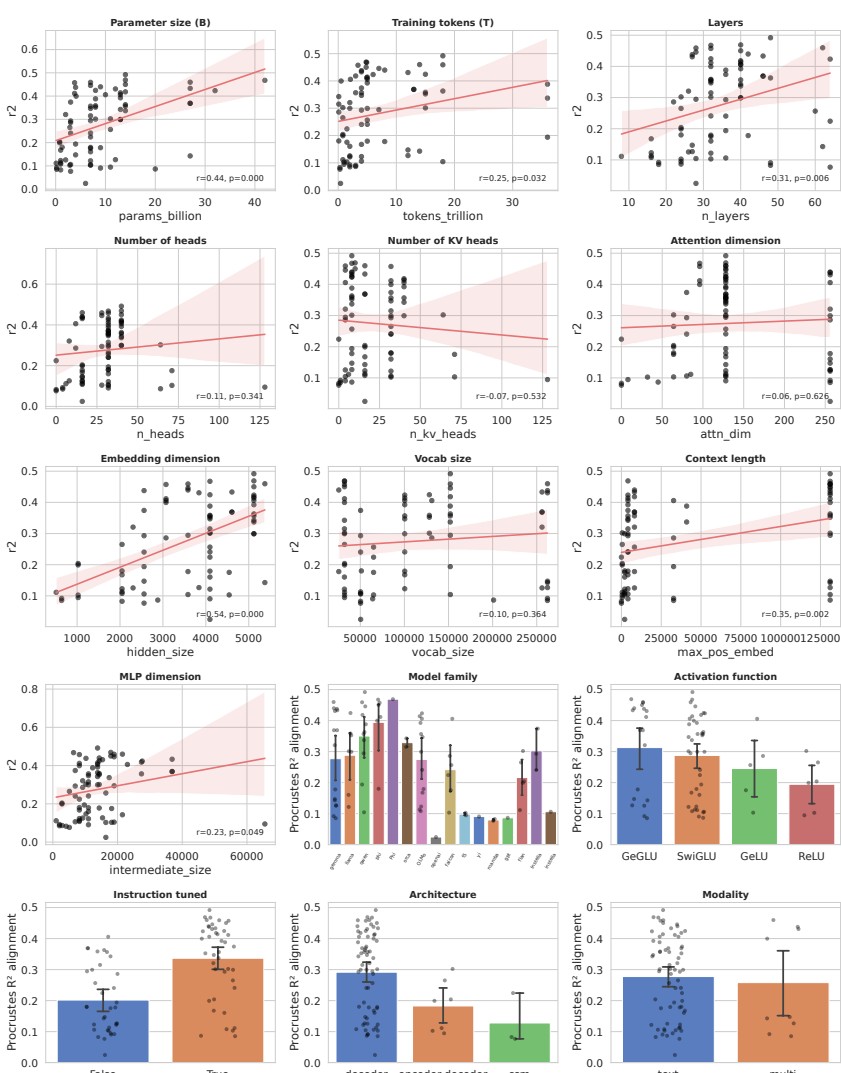

Figure 6: Categorical and continuous between all computational ingredients and human-model alignment. All significant effects in the full mixed linear-model are reported in the main text, Figure 2.

## A.10  *t*-SNE VISUALIZATIONS

Table 4: Complete ranking of models by R² score

| Rank | Model Name | R² Score | Rank | Model Name | R² Score |
|------|-----------|----------|------|-----------|----------|
| 1 | qwen25-14b-instruct | 0.492 | 39 | qwen2-7b | 0.294 |
| 2 | Phi-3-medium-4k-instruct | 0.469 | 40 | Instella-3B-SFT | 0.293 |
| 3 | Phi-3.5-MoE-instruct | 0.468 | 41 | Falcon3-3B-Base | 0.286 |
| 4 | gemma-3-27b-it | 0.460 | 42 | flan-t5-xl | 0.265 |
| 5 | qwen25-7b-instruct | 0.459 | 43 | falcon-11b | 0.257 |
| 6 | Phi-3.5-mini-instruct | 0.457 | 44 | OLMo-2-1124-7B-Instruct | 0.241 |
| 7 | phi-3-medium-128k-instruct | 0.450 | 45 | Instella-3B | 0.241 |
| 8 | qwen2-7b-instruct | 0.446 | 46 | Falcon3-Mamba-7B-Base | 0.224 |
| 9 | gemma-2-9b-it | 0.440 | 47 | flan-t5-large | 0.199 |
| 10 | gemma-3-4b-it | 0.438 | 48 | Qwen3-4B-Base | 0.194 |
| 11 | gemma-2-27b-it | 0.433 | 49 | phi-1_5 | 0.180 |
| 12 | gemma-3-12b-it | 0.431 | 50 | OLMo-2-1124-7B | 0.179 |
| 13 | Llama-3.1-8B-Instruct | 0.424 | 51 | gemma-2-9b | 0.178 |
| 14 | OLMo-2-0325-32B-Instruct | 0.423 | 52 | falcon-7b | 0.175 |
| 15 | qwen-14b-chat | 0.418 | 53 | OLMo-2-0425-1B-SFT | 0.168 |
| 16 | OLMo-2-1124-13B-Instruct | 0.414 | 54 | llama2-7b-chat | 0.160 |
| 17 | phi-3-mini-128k-instruct | 0.412 | 55 | gemma-3-4b-pt | 0.147 |
| 18 | OLMo-2-1124-13B-DPO | 0.406 | 56 | gemma-3-27b-pt | 0.143 |
| 19 | Falcon3-10B-Base | 0.406 | 57 | gemma-3-12b-pt | 0.127 |
| 20 | Phi-3.5-vision-instruct | 0.400 | 58 | gemma-2-2b | 0.126 |
| 21 | OLMo-2-1124-13B-SFT | 0.392 | 59 | OLMo-2-0425-1B | 0.123 |
| 22 | qwen3-8b | 0.388 | 60 | llama2-7b | 0.122 |
| 23 | Instella-3B-Instruct | 0.374 | 61 | AMD-OLMo-1B | 0.113 |
| 24 | gemma-2-27b | 0.369 | 62 | flan-t5-small | 0.111 |
| 25 | qwen25-14b | 0.363 | 63 | AMD-OLMo-1B-SFT-DPO | 0.108 |
| 26 | Llama-3.1-Tulu-3-8B | 0.358 | 64 | Instella-3B-Stage1 | 0.106 |
| 27 | qwen-14b | 0.357 | 65 | qwen25-7b | 0.104 |
| 28 | OLMo-2-1124-7B-DPO | 0.357 | 66 | falcon-7b-instruct | 0.103 |
| 29 | Llama-3.1-Tulu-3-8B-DPO | 0.352 | 67 | t5-3b | 0.103 |
| 30 | OLMo-2-1124-7B-SFT | 0.347 | 68 | t5-11b | 0.095 |
| 31 | orca-2-13b | 0.343 | 69 | gemma-3-270m | 0.092 |
| 32 | qwen3-14b | 0.337 | 70 | Yi-9B | 0.091 |
| 33 | gemma-2-2b-it | 0.321 | 71 | gpt-oss-20b | 0.087 |
| 34 | orca-2-7b | 0.315 | 72 | gemma-3-270m-it | 0.085 |
| 35 | flan-t5-xxl | 0.302 | 73 | mamba-1.4b-hf | 0.083 |
| 36 | Llama-3.1-Tulu-3-8B-SFT | 0.301 | 74 | mamba-2.8b-hf | 0.077 |
| 37 | llama2-13b-chat | 0.299 | 75 | gpt-j-6b | 0.025 |
| 38 | OLMo-2-1124-13B | 0.299 | | | |

Figure 7: t-SNE visualizations of model embeddings (Part 1). $R^2$ scores are in parentheses.

(1) Qwen2.5-14B-Inst (0.492)  (2) Phi-3-Med-4k (0.469)  (3) Phi-3.5-MoE (0.468)

(4) Gemma-3-27B-IT (0.460)  (5) Qwen2.5-7B-Inst (0.459)  (6) Phi-3.5-Mini (0.457)

(7) Phi-3-Med-128k (0.450)  (8) Qwen2-7B-Inst (0.446)  (9) Gemma-2-9B-IT (0.440)

(10) Gemma-3-4B-IT (0.438)  (11) Gemma-2-27B-IT (0.433)  (12) Gemma-3-12B-IT (0.431)

Figure 8: t-SNE visualizations of model embeddings (Part 2). R² scores are in parentheses.

(13) Llama-3.1-8B (0.424)

(14) OLMo-2-32B (0.423)

(15) Qwen-14B-Chat (0.418)

(16) OLMo-2-13B-Inst (0.414)

(17) Phi-3-Mini-128k (0.412)

(18) OLMo-2-13B-DPO (0.406)

(19) Falcon3-10B (0.406)

(20) Phi-3.5-Vision (0.400)

(21) OLMo-2-13B-SFT (0.392)

(22) Qwen3-8B (0.388)

(23) Instella-3B-Inst (0.374)

(24) Gemma-2-27B (0.369)

Figure 9: t-SNE visualizations of model embeddings (Part 3). $R^2$ scores are in parentheses.

(25) Qwen2.5-14B (0.363)

(26) Llama-3.1-Tulu (0.358)

(27) Qwen-14B (0.357)

(28) OLMo-2-7B-DPO (0.357)

(29) Llama-3.1-Tulu-DPO (0.352)

(30) OLMo-2-7B-SFT (0.347)

(31) Orca-2-13B (0.343)

(32) Qwen3-14B (0.337)

(33) Gemma-2-2B-IT (0.321)

(34) Orca-2-7B (0.315)

(35) FLAN-T5-XXL (0.302)

(36) Llama-3.1-Tulu-SFT (0.301)

Figure 10: t-SNE visualizations of model embeddings (Part 4). R² scores are in parentheses.

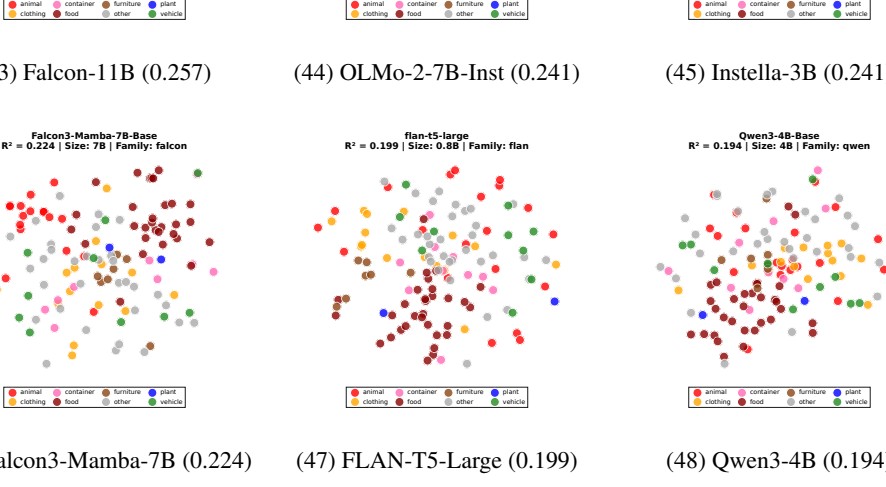

(37) Llama2-13B-Chat (0.299)   (38) OLMo-2-13B (0.299)   (39) Qwen2-7B (0.294)

(40) Instella-3B-SFT (0.293)   (41) Falcon3-3B (0.286)   (42) FLAN-T5-XL (0.265)

(43) Falcon-11B (0.257)   (44) OLMo-2-7B-Inst (0.241)   (45) Instella-3B (0.241)

(46) Falcon3-Mamba-7B (0.224)   (47) FLAN-T5-Large (0.199)   (48) Qwen3-4B (0.194)

Figure 11: t-SNE visualizations of model embeddings (Part 5). $R^2$ scores are in parentheses.

(49) Phi-1.5 (0.180)  (50) OLMo-2-7B (0.179)  (51) Gemma-2-9B (0.178)

(52) Falcon-7B (0.175)  (53) OLMo-2-1B-SFT (0.168)  (54) Llama2-7B-Chat (0.160)

(55) Gemma-3-4B-PT (0.147)  (56) Gemma-3-27B-PT (0.143)  (57) Gemma-3-12B-PT (0.127)

(58) Gemma-2-2B (0.126)  (59) OLMo-2-1B (0.123)  (60) Llama2-7B (0.122)

Figure 12: t-SNE visualizations of model embeddings (Part 6). R² scores are in parentheses.

(61) AMD-OLMo-1B (0.113)

(62) FLAN-T5-Small (0.111)

(63) AMD-OLMo-1B-DPO (0.108)

(64) Instella-3B-Stage1 (0.106)

(65) Qwen2.5-7B (0.104)

(66) Falcon-7B-Inst (0.103)

(67) T5-3B (0.103)

(68) T5-11B (0.095)

(69) Gemma-3-270M (0.092)

(70) Yi-9B (0.091)

(71) GPT-OSS-20B (0.087)

(72) Gemma-3-270M-IT (0.085)

Figure 13: t-SNE visualizations of model embeddings (Part 7). R² scores are in parentheses.

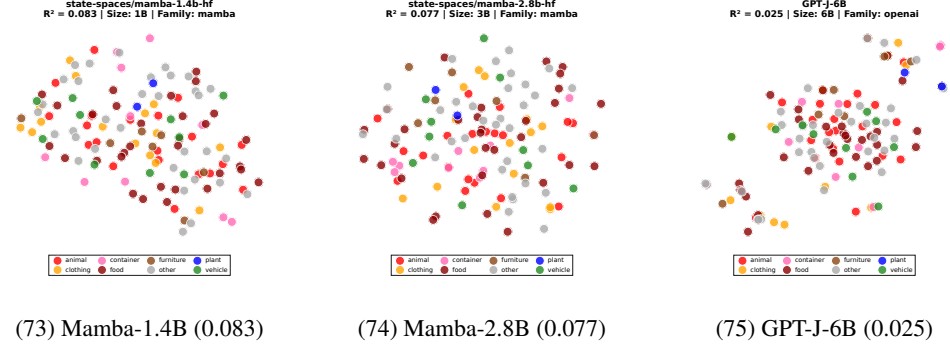

(73) Mamba-1.4B (0.083)   (74) Mamba-2.8B (0.077)   (75) GPT-J-6B (0.025)

