# OpenReview forum: "Uncovering the computational ingredients that support human-like conceptual representations in large language models"
_ICLR.cc/2026/Conference — ICLR 2026 Conference Desk Rejected Submission_

### Official Review · Reviewer_7rre · 2025-10-29

**Soundness:** 3
**Presentation:** 3
**Contribution:** 2
**Rating:** 4
**Confidence:** 4

**Summary:**

The paper studies the current state of LLMs as computational cognitive models form the representational perspective. The paper specifically focuses on identifying the computational ingredients that would bias models towards more human-like representations. The paper focuses on the THINGS dataset, a dataset that contains human similarity judgements for 4.7 million triplets, and compares subsets thereof to a suite of 77 foundation models.

**Strengths:**

- The paper is overall very well-written and clearly describes its approach, results and interpretation.
- The empirical results are extensive regarding number of models and downstream benchmarking datasets.
- The analyses are novel to my knowledge, i.e., the relationship between alignment and performance on standard LLM benchmarks.
- The paper includes an honest and balanced discussion and limitations sections. The results appear reproducible and methods are clearly described.

**Weaknesses:**

- The paper presents an empirical study focused on a single dataset (THINGS).
- The overall analyses unfortunately remained focused on dataset-wide reporting of alignment scores. I think the paper would benefit from a more detailed analysis of alignment results in dependence of dataset composition, at least for a subset, e.g. particularly well-aligned, models. Relevant dimensions could include triplet term-similarity levels, concreteness or linguistically frequent tokens (see the mentioned ecosat data).
- Similarly, the overall alignment of models as measured via $R^2$ is at around 0.5 for the most human-like models, the paper unfortunately does not present an analysis of error modes and shortcomings/limits of current models. What are the, e.g. semantic, types or families of triplets that models struggle with w.r.t. alignment? The THINGS dataset contains additional annotations, e.g. “core object categories,” “animacy” and “size ratings” that would be useful to better understand the observed limits.



**Additional Refs**

- Effect of instruction-tuning on language Models and conceptual alignment:
    - [Lin25] Linhardt, L., Neuhäuser, T., Tětková, L., & Eberle, O. Cat, Rat, Meow: On the Alignment of Language Model and Human Term-   Similarity Judgments. In Second Workshop on Representational Alignment at ICLR 2025.
    - [Aw24] Aw, K. L., Montariol, S., AlKhamissi, B., Schrimpf, M., & Bosselut, A. Instruction-tuning aligns llms to the human brain. COLM 2024.

**Questions:**

- What is the purpose of using the ordinal embedding algorithm to fit an embedding model to judgement data over directly using the raw model embeddings, e.g. that of the answer token? Is the main reason to match the dimensionality of human-based embeddings?
- Did you observe any relevant differences in representational geometry (see also points made above)?

**Details Of Ethics Concerns:**

None.

---

> ### Author Response · Authors · 2025-11-25
> **Response to Reviewer 7rre's noted weaknesses**
>
> We thank the reviewer for their thoughtful comments and for their suggestions, which we feel have helped strengthen the paper.
> We first address the main points of weakness noted and then respond to the questions.
>
> Regarding the first weakness (**indexing results solely on THINGS**). While it is true that THINGS is a single dataset, the scale and diversity of the dataset paired with the richness of metadata associated with it makes it a uniquely well-suited dataset for studying semantic alignment. However, based on similar feedback from reviewer Gksp we have now included a new analysis of abstract emotion terms. We duplicate our response below –
> >We do agree that it would be useful to know how our findings generalize to abstract concepts.
>
> >In order to help answer this, we computed semantic embeddings for 135 abstract emotion concepts  (Shaver et al., 1987)using the same workflow described in the paper’s current methods section. We then estimated semantic embeddings for our language models by having them perform over 15,000 triplet judgments per model (we needed fewer judgments here because we have fewer concepts). Due to current space constraints we report the results here, but if accepted we will use the new page to report on this analysis and the other analyses in response to the third weakness.
>
> >As of this response, we have results from 37/77 of our models (due to compute constraints). However, we are finding large concordances with our findings from concrete concepts. See (anonymized) graph below-
> https://imgur.com/gallery/abstract-concepts-okJ7qia
>
> >While the trends for the predictors are similar to what we’ve found for concrete concepts, the effect of instruction tuning and attention dimensionality are also already statistically significant. We do note that ‘number of layers’ is significant currently, which was not the case for concrete concepts, but this is likely because we do not have enough statistical power to estimate the other effects, due to only running these regressions on a reduced model set.
> However, regardless of if the current pattern of significance holds or changes as the remaining models complete running, we believe the addition of this analysis section will show the reader the extent to which there are concordances or divergences between abstract and concrete concepts, addressing the core question raised by the reviewer.
>
> >Shaver, P., Schwartz, J., Kirson, D., & O'connor, C. (1987). Emotion knowledge: further exploration of a prototype approach. Journal of personality and social psychology, 52(6), 1061.
>
> ---

---

> ### Author Response · Authors · 2025-11-25
> **Response to Reviewer 7rre's noted weaknesses (continued)**
>
> ---
>
> Weaknesses 2 and 3 are shortcomings of the original submission we take quite seriously. We have attempted to address these limitations with a new set of analyses included in our response to reviewer Gksp. For convenience we duplicate our description below –
>
> >Our focus with these new analyses was to give the reader an insight into _what_ concepts or what properties of concepts were predictive of alignment. Due to the 8-page limit we do not include these in the edited pdf yet, but if accepted, would include this using the extra 9th page. We still report stats and (anonymized) links to new figures below.
>
> >We first considered whether different superordinate categories of concepts showed different degrees of alignment. Specifically, we grouped each of our 128 concepts into the following higher order categories - Furniture, Container, Vehicle, Clothing, Food, Animal, and Other. We next computed the average Procrustes $R^2$ within each of these categories for each model and compared the alignment across categories.
> We found the highest degree of alignment was for inanimate objects like containers and furniture and the least aligned concepts were those that were food and animals. See graph here - https://imgur.com/gallery/model-alignment-per-concept-category-AyOUD7C
>
> >We further found that models regardless of model size (here grouped into four representative bins) were somewhat correlated on which concept categories were most-to-least aligned (Kendall’s $\tau$ = .45.
> We next leveraged metadata about each concept from THINGS including their memorability, typicality, and concreteness. For each of our concepts, we computed the cosine distance between the true human embedding and the language model response-based embedding. This gives us a per item measure of alignment. We then fit mixed-effects linear regression models predicting the cosine distance from a set of factors shown in the following figure. The figure plots the $t$-scores of each predictor indicating the amount of variance uniquely explained by that predictor. Lower bars indicate predictors that were anti-correlated with cosine distance, which implies higher alignment.  –
> . https://imgur.com/gallery/things-alignment-bJrDstK
>
> >We generally found that concepts that were generally more familiar and typical of their parent category (to humans) were more strongly aligned. More memorable and more concrete concepts were also more aligned. Lastly, concepts that were polysemous (can have many meanings) were least aligned, which makes sense since it is difficult to align concepts that are broad in their semantic coverage.
> Overall, we believe that this pair of analyses strongly adds greater insight into the kinds of cognitive factors that correspond to greater or lesser alignment for some concepts over others and adds more qualitative insights as well. We hope this addresses the weakness raised by the reviewer.
> NOTE: We did not include the regression tables here due to not clutter the response, but we are happy to provide if it will be helpful
>
> Lastly, thank you for the helpful references. We will include both these relevant papers in our amended introduction/background.
>
> ---
> ---

---

> ### Author Response · Authors · 2025-11-25
> **Response to Reviewer 7rre's questions**
>
> In terms of Questions raised -
>
> **Q1**. Yes, we chose to keep the behavioral task as similar as possible for models and humans to facilitate a fair apples-to-apples comparison between these two systems. As the reviewer might be suggesting, there is always the possibility of probing model activations directly to estimate model representations. We chose not to pursue this path for several principled reasons. (1) There is no single method for estimating representations for single word concepts that has been shown to be better than the rest, thus introducing another degree of freedom, (2) similar to adjacent work in sparse autoencoders, the specific dataset of tokens used to probe for concept representations might strongly influence the nature of representations surfaced, without strong theory linking how token distribution maps onto concept representation variability, and (3) native model embeddings would be extremely high dimensional requiring decisions to be made on how to reduce the dimensionality to match human responses (or to solely rely on representational similarity analysis (RSA) as a measure).
>
> ---
>
> **Q2**. Thank you for this question! For a qualitative sense of how representational geometry of concepts varied across models, we encourage the reviewer to view the t-SNE plots included in the appendix (Figures 7-13).
> For a more quantitative assessment of how representational geometry between humans and models on a concept-by-concept level, we refer the reviewer to the new analyses we report above in response to weaknesses 2 and 3. We believe the new _concept-by-concept_ analysis of how alignment varies addresses core aspects of this question.
>
> ---
> ---

---

> > ### Comment · Reviewer_7rre · 2025-11-27
> >
> > Thank you for effectively addressing my concerns and answering the raised questions. I think the revised paper is clearly improved and I have raised my score from 4 to 6.

---

> > > ### Author Response · Authors · 2025-11-27
> > > **Response to Reviewer 7rre**
> > >
> > > We once again thank the reviewer for their helpful feedback, which helped improve the paper, and are grateful for the increased score.

---

### Official Review · Reviewer_Gksp · 2025-11-01

**Soundness:** 3
**Presentation:** 3
**Contribution:** 3
**Rating:** 8
**Confidence:** 3

**Summary:**

This paper investigates which computational ingredients, ranging from architecture, size, fine-tuning, data size, and training regimes, could predict human-like representational alignment in large language models (LLMs). Using over 77 open-weight models spanning major model families (Llama, Qwen, Gemma, Phi, OLMo, Falcon, etc.), the authors applied a triadic similarity judgment task based on 128 object concepts from the THINGS database, which includes >4.7M human triplet judgments. Each model completes 35,000 trials of the same task (“Which of y/z is most similar to x?”), and the authors construct model semantic embeddings using ordinal embedding techniques comparable to the SPoSE embeddings from human data. By computing representational alignment (Procrustes R²) between model and human embeddings and relate alignment to architectural and training factors. The found that Instruction fine-tuning is the strongest predictor of human-model alignment; embedding and MLP dimensionality positively correlate with alignment; model scale and training data amount contribute moderately. Multimodal pretraining, activation function, and vocabulary size have little or negative impact.

The authors conclude that instruction tuning and model representational capacity are the key computational ingredients of human-like conceptual structure, and that human-based representational benchmarks provide unique insights beyond standard NLP tasks.

**Strengths:**

1. Ambitious and comprehensive analysis: Evaluating 77 LLMs across diverse families and parameterizations is an impressive effort.
2. Conceptually grounded approach: The use of the THINGS dataset and triplet similarity judgments directly connects AI benchmarking with established cognitive science methods.
3. Clear and interpretable results: The paper identifies consistent, actionable findings (e.g., importance of instruction tuning and representational dimensionality) that have implications for both model design and theories of human cognition.
4. Bridging cognitive science and AI: It positions representational alignment as a measurable axis of “human-likeness” in LLMs, helping to bridge psychological theories of conceptual structure with LLM evaluation.

**Weaknesses:**

1. The THINGS dataset focuses on concrete object concepts. It's unclear whether the findings generalize to abstract or relational knowledge, which is central to many cognitive domains.
2. The study establishes associations but not causal explanations for why instruction tuning or architectural dimensionality yield better human alignment. Simulated ablations or controlled fine-tuning experiments could strengthen the causal claims.
3. While alignment scores and t-SNE maps are presented, the paper does not analyze which semantic dimensions (e.g., animacy, size, tool–animal axes) drive human-model similarity. This limits its contribution to understanding the nature of human-like representations.

**Questions:**

1. The human embeddings are taken from SPoSE, while model embeddings are constructed using an ordinal embedding algorithm. Given that both humans and models performed the same triplet-similarity task, could SPoSE be applied to the model triplet data as well? Using the same embedding algorithm might yield a more symmetric comparison and reduce potential geometric biases introduced by differing regularization assumptions.

2. The paper states that both human and model embeddings are reduced to ≈29 dimensions to allow Procrustes alignment. It's unclear whether this dimensionality was fixed to match human SPoSE variance explained, or optimized separately for each model? Would the main results hold if the model embeddings used their optimal dimensionality rather than being forced to match the human space?

3. While R² quantifies overall alignment, it would be valuable to understand which semantic dimensions (e.g., animacy, toolness, color) drive the alignment between humans and models. Did the authors attempt to interpret or visualize specific embedding dimensions after alignment?

4. Since the THINGS dataset focuses on concrete visual object concepts, do the observed effects of instruction tuning and architectural dimensionality generalize to more abstract, relational, or linguistic domains?

5. The authors mention collecting 35 000 triplets per model. How sensitive are the alignment results to the number of triplets? Was the choice of ordinal embedding partly motivated by data efficiency compared to SPoSE?

---

> ### Author Response · Authors · 2025-11-25
> **Response to Reviewer Gksp's noted weaknesses**
>
> We thank the reviewer for their close reading of our paper and for raising several helpful points, many of which we have incorporated into our paper as new analyses or modifications to text. We respond point-by-point below.
>
> Regarding the first weakness (**THINGS being restricted to mostly concrete concepts**): While we believe that THINGS is one-of-a-kind dataset in terms of its semantic diversity and richness in cognitive metadata, we do agree that it would be useful to know how our findings generalize to abstract concepts.
>
> In order to help answer this, we computed semantic embeddings for 135 abstract emotion (Shaver et al., 1987) concepts using the same workflow described in the paper’s current methods section. We then estimated semantic embeddings for our language models by having them perform over 15,000 triplet judgments per model (we needed fewer judgments here because we have fewer concepts). Due to current space constraints we report the results here, but if accepted we will use the new page to report on this analysis and the other analyses in response to the third weakness.
>
> As of this response, we have results from 37/77 of our models (due to compute constraints). However, we are finding large concordances with our findings from concrete concepts. See (anonymized) graph below-
> https://imgur.com/gallery/abstract-concepts-okJ7qia
> While the trends for the predictors are similar to what we’ve found for concrete concepts, the effect of instruction tuning and attention dimensionality are also already statistically significant. We do note that ‘number of layers’ is significant currently, which was not the case for concrete concepts, but this is likely because we do not have enough statistical power to estimate the other effects, due to only running these regressions on a reduced model set.
> However, regardless of if the current pattern of significance holds or changes as the remaining models complete running, we believe the addition of this analysis section will show the reader the extent to which there are concordances or divergences between abstract and concrete concepts, addressing the core question raised by the reviewer.
>
> References:
>
> Shaver, P., Schwartz, J., Kirson, D., & O'connor, C. (1987). Emotion knowledge: further exploration of a prototype approach. Journal of personality and social psychology, 52(6), 1061.
>
> ---
>
> Regarding the second weakness (**limited causal takeaways**): We agree with the general point that our analyses do not test for causal relationships between the factors investigated and alignment. As the reviewer notes, careful controlled rearing experiments would be needed where models are trained under different configurations to make strong causal claims. Nevertheless, we believe that our analyses (across many models) is foundational for (1) uncovering the set of factors that might be potentially useful for said controlled rearing experiments and (2) As noted in our response to reviewer 7rre, we use rigorous statistical models that are foundational for making inferences about which properties influence behavior in the cognitive sciences.
>
> ---

---

> ### Author Response · Authors · 2025-11-25
> **Response to Reviewer Gksp's noted weaknesses (continued.)**
>
> Regarding the third weakness (**analysis of semantic dimensions**): We agree that the lack of such analyses was a shortcoming in our initial submission and we have now made a concerted effort to address this. We describe a new set of analyses, which we aim to include in the revised paper, below –
>
> Our focus with these new analyses was to give the reader an insight into _what_ concepts or what properties of concepts were predictive of alignment. Due to the 8-page limit we do not include these in the edited pdf yet, but if accepted, would include this using the extra 9th page. We still report stats and (anonymized) links to new figures below.
>
> We first considered whether different superordinate categories of concepts showed different degrees of alignment. Specifically, we grouped each of our 128 concepts into the following higher order categories - Furniture, Container, Vehicle, Clothing, Food, Animal, and Other. We next computed the average Procrustes $R^2$ within each of these categories for each model and compared the alignment across categories.
> We found the highest degree of alignment was for inanimate objects like containers and furniture and the least aligned concepts were those that were food and animals. See graph here - https://imgur.com/gallery/model-alignment-per-concept-category-AyOUD7C
>
> We further found that models regardless of model size (here grouped into four representative bins) were somewhat correlated on which concept categories were most-to-least aligned (Kendall’s $\tau$ = .45).
> We next leveraged metadata about each concept from THINGS including their memorability, typicality, and concreteness. For each of our concepts, we computed the cosine distance between the true human embedding and the language model response-based embedding. This gives us a per item measure of alignment. We then fit mixed-effects linear regression models predicting the cosine distance from a set of factors shown in the following figure. The figure plots the $t$-scores of each predictor indicating the amount of variance uniquely explained by that predictor. Lower bars indicate predictors that were anti-correlated with cosine distance, which implies higher alignment  – https://imgur.com/gallery/things-alignment-bJrDstK
>
> We generally found that concepts that were generally more familiar and typical of their parent category (to humans) were more strongly aligned. More memorable and more concrete concepts were also more aligned. Lastly, concepts that were polysemous (can have many meanings) were least aligned, which makes sense since it is difficult to align concepts that are broad in their semantic coverage.
> Overall, we believe that this pair of analyses strongly adds greater insight into the kinds of cognitive factors that correspond to greater or lesser alignment for some concepts over others and adds more qualitative insights as well. We hope this addresses the weakness raised by the reviewer.
> NOTE: We did not include the regression tables here to not clutter the response, but we are happy to provide if it will be helpful
>
> ---
> ---

---

> > ### Author Response · Authors · 2025-11-25
> > **Response to Reviewer Gksp's questions**
> >
> > In terms of Questions raised -
> >
> > **Q1.** Thank you for raising this point. We specifically chose the crowd kernel based approach to estimating embeddings due to rigorous analytical work (Sievert et al., 2023) providing strong theoretical bounds on how the number of samples relate to embedding quality. We were unable to find similar guarantees for the SPoSE algorithm, making it unclear how many samples it would take to arrive at ‘good’ embeddings/ Given that we needed to estimate embeddings for each model, we chose this approach (which involved collecting data on which of two options was most similar to the target, which is subtly different from the odd-one-out task) for efficiency without needing to compromise on quality. However, the reviewer’s main concern is a valid one so now we will now include a figure in the Appendix showing that the SpoSe embeddings and Crowd Kernel based embeddings are highly correlated, indicating that these methods are highly consistent in their yielded representational geometry.
> >
> > Notably, we found an extremely high Procrustes $R^2$ of 0.99 between embeddings estimated using SPoSE and the crowd-kernel algorithm for a set of 77 concepts for which we had human anchored similarity judgments (which the latter algorithm expects as input). This value is essentially a measure of human-human reliability, which should be high if the algorithms are estimating similar dimensions. We use this as our basis for licensing comparisons between embeddings estimated by both algorithm. The reviewer can refer to the similarity structure being expressed by both algorithms as being similar in this (anonymized) image - https://imgur.com/gallery/spose-vs-salmon-AfOM7uV
> >
> > ---
> >
> > **Q2.** We apologize for failing to motivate this choice of dimensionality. Yes, we did choose our dimensionality to match the dimensionality that explained over 95% variance in the human embeddings. However, we also believe the question about how robust our results are to dimensionality are important. To test the robustness of our findings, we estimated embeddings using 20-30 dimensions for 3 models broadly representing small, medium, and large models in our set (gemma-3 4B, 12B, and 27B; see anonymized figure here - https://imgur.com/gallery/loss-validation-accuracy-WqtPwJQ). We found that tracking both the validation accuracy on held out triplets and on the crowd kernel loss that embeddings were highly similar in this dimensionality range indicating that models would have also aligned with human embeddings at lower dimensionalities; however maintaining dimensionality between the agent types (model and LLM) allows for computing the dimension-matched Procrustes $R^2$. We will include these results in the Appendix for the interested reader.
> >
> > ---
> >
> > **Q3.** This is related to the third weakness, and we refer the reviewer to our response above.
> >
> > ---
> >
> > **Q4.** This is related to the first weakness and we refer the reviewer to our response above.
> >
> > ---
> >
> > **Q5.** There are theoretical guarantees from Sievert et al. (2023) on the quality of embeddings as a function of the number of samples. As noted in the paper we require at least $ndlog_2(n)$ triplet trials to estimate a reliable embedding, where n is the number of items (here 128), d is the estimated dimensionality (here 30, based on the human data). This equation tells us we need at least 26,880 trials. Usually, it is recommended to collect more than the minimum to account for noise in the judgements. Based on our compute budget, we estimated that we could collect 35,000 judgments per model. Thus, to answer the main question, collecting fewer than 26k trials would lead to noisier embeddings, which would lead to poorer alignment. Here, we present embeddings based on the 35k trials to give models their ‘best shot’ at matching human performance. Lastly, yes, we did choose the ordinal embedding algorithm for efficiency reasons.
> >
> > References:
> >
> > Sievert, S., Nowak, R., & Rogers, T. T. (2023). Efficiently learning relative similarity embeddings with crowdsourcing. Journal of open source software, 8(84).

---

### Official Review · Reviewer_k7pP · 2025-11-01

**Soundness:** 3
**Presentation:** 3
**Contribution:** 4
**Rating:** 8
**Confidence:** 4

**Summary:**

This paper provides an in-depth analysis of concept representations in LLMs and seeks to discover the alignment of models with human representations of concepts.  The findings are informative in uncovering how alignment arises, e.g., increasingly during post-training stages, understanding the amount of context required to achieve alignment, and understanding where in the architecture concepts best align.

**Strengths:**

- The analysis is thorough and the findings are interesting and informative.
- Understanding concept representations is an interesting topic and could help to unlock robustness issues with models, e.g., why models fail in particular ways.
- The authors acknowledge that the distribution of models is skewed to particular model classes, but they have considered a large number of models.

**Weaknesses:**

- I do realize this cannot be addressed by the authors, but it is hard to isolate individual aspects of data/model size/architecture/etc. to hone in on specific aspects that might be responsible for more/less alignment.

**Questions:**

- Line 055: should data be in the list of ingredients? I realize that the list is not exhaustive, but data feels an important component.
- Line 077: “the geometry human” > “the geometry of human”.
- Line 113: There is an extraneous space after “error-bounds”.
- Line 228: "The incorrect opening quotes are used.
- Line 259: Should “QUESTION: Which item  ...” be in quotes?
- Line 279: “Procrustes transforms find” > “Procrustes transform finds”.
- Figure 3: use colors that better distinguish the lines.
- Figure 3: there seems to be spacing issues between the main text and the caption — the spacing makes it feel cramped.

---

> ### Author Response · Authors · 2025-11-25
> **Response to Reviewer k7pP**
>
> We thank the reviewer for their review and thoughtful comments.
> Regarding the main weakness raised -
> >I do realize this cannot be addressed by the authors, but it is hard to isolate individual aspects of data/model size/architecture/etc. to hone in on specific aspects that might be responsible for more/less alignment.
>
> We agree that the current analytical approach will not help resolve the causal role of the individual factors tested in training models that are aligned conceptually with humans.
> However, we do wish to emphasize that our analyses, particularly those under Section 4 (Which computational ingredients have the greatest relative contribution towards human-model alignment?), do leverage the best-in-class statistical analytical methods that allow us to perform credit assignment on different factors that might lead to higher/lower alignment.
> Specifically, the mixed-effects models that we used surface fitted parameters that indicate how well a given factor is predictive of alignment, controlling for all other factors.
> To help make this more clear, we have added a new sentence to the text around Line 359 to include
> >“ … unique variance after others are considered. One way to estimate the causal role of each ingredient is to exert fine-grained control over model parameters and train many language models from scratch and test for the alignment between the resulting models and humans. Given the extreme cost of training so many models (especially those are of scale 20B+, we instead opted to use the best-in-class statistical inference models to understand which factors are more important relative to the rest. Specifically, we used mixed-effects multiple regression model (Lindstrom & Bates, 1990) predicting Procrustes R2 from all computational ingredients. Mixed-effects models combine the interpretability of OLS with random effects to capture item-level variance (Barr et al., 2013).”
>
> Additionally, we will also include the full table of regression coefficients in the appendix. This helps convey to the reader not only which factors are significantly predictive of alignment but by how much (beta weights).
>
> ---
>
> We will address the typographical errors noted in the questions by the reviewer, which we are deeply grateful for.

---

### Author Response · Authors · 2025-12-02
**Summary of Changes**

Firstly, we want to once again thank the reviewers who raised several useful points of feedback, many of which have inspired new analyses and changes, which we summarize below.
We appreciate the close reading of our manuscript and their noting of points that could use more clarification. We have included several new plots using anonymized links in the individual reviewer responses, and have noted details which we will include in both the appendix and the extra page, if the paper were to be accepted. We hope our changes have addressed the key concerns and limitations.

* **New Analyses generalizing findings to abstract an domain (emotions) in response to limitation of the THINGS database containing only concrete concepts**: We now report analyses on a new experiment using 135 abstract emotion concepts. We found that our finding findings regarding the importance of instruction tuning and dimensionality generalize well to this new domain of concepts. We also note some interesting differences with respect to the original analyses on concrete concepts (noted in the response to the reviewer). We will include these findings in the main text using the additional extra page, if the paper were to be accepted (Reviewers Gksp and 7rre).
* **Fine-grained concept-level analysis of (mis)alignment**: In response to reviewers asking for further insight into what properties of concepts lead to alignment (or lack thereof), we have now gone beyond dataset-wide metrics with two key sets of analyses:
    * **Semantic Category-level alignment analyses**: we found that models align best with inanimate objects (e.g., containers, furniture) and worst with animate concepts (e.g., animals)
    * **Regression models that model concept-level alignment as a function of concept metadata**: Using the metadata that comes with THINGS, we found that concept familiarity, typicality, and concreteness are positive predictors of alignment, while polysemy is a negative predictor of alignment. These two analyses help address key issues raised by both Reviewers Gksp and 7rre.


* **Methodological Robustness & Validation**:
We will add Appendix figures to validate our embedding choices, which we briefly recap here and then in more detail under specific reviewer comments:
    * We demonstrated that our crowd-kernel embeddings are highly consistent with SPoSE embeddings ($R^2 = 0.99$), justifying our choice of embedding algorithm (Reviewer Gksp).

    * We verified that our results are robust to variations in embedding dimensionality (20–30 dims) across model sizes (Reviewer Gksp).

* **Statistical Interpretation & Causality**: We further refined the main text to clarify the "credit assignment" role of our mixed-effects models in isolating unique variance, while acknowledging that stronger causal claims cannot be made without training many models from scratch while varying different computational ingredients. We will also add full regression coefficient tables to the Appendix for transparency (Reviewers k7pP and Gksp).

* **Justification of Behavioral Task**: We clarified our choice to use behavioral triplet tasks rather than internal probing, emphasizing the necessity of an "apples-to-apples" comparison with human data and the avoidance of biases introduced by specific probing datasets and also emphasizing how triplet tasks expose representations used for a variety of semantic tasks in humans. (Reviewer 7rre).
* **Corrections**: We corrected the typographical errors and figure formatting issues identified (Reviewer k7pP).

---

### Note · Program_Chairs · 2026-01-17
**Submission Desk Rejected by Program Chairs**

The following references in this submission do not refer to real documents and/or have major errors in bibliographic information:

     Sishuo Han, Wenshan Cheng, Zhiruo Liu, Kaixin Shi, Jing Zhou, Xiang Ren, and Yizhou Wang. MUSR: A multi-step unsupervised scientific reasoning benchmark. arXiv preprint arXiv:2405.08726, 2024.
    Joosung Nam, Juyeon Lee, Dongkeun Chung, Juyoung Song, Sangwoo Yoon, Sungjin Jung, Seungjong Min, and Yongjin Choi. Idealignbench: Measuring and improving value alignment of large language models. arXiv preprint arXiv:2412.03924, 2024.